# Comparative single-cell profiling reveals distinct cardiac resident macrophages essential for zebrafish heart regeneration

Ke-Hsuan Wei[1,2], I-Ting Lin[2], Kaushik Chowdhury[2,3], Khai Lone Lim[2,3], Kuan-Ting Liu[4], Tai-Ming Ko[2,4], Yao-Ming Chang[2], Kai-Chien Yang[2,5], Shih-Lei (Ben) Lai[1,2,3]*

[1]Graduate Institute of Life Sciences, National Defense Medical Center, Taipei, Taiwan; [2]Institute of Biomedical Sciences, Academia Sinica, Taipei, Taiwan; [3]Taiwan International Graduate Program in Molecular Medicine, National Yang Ming Chiao Tung University, Taipei, Taiwan; [4]Department of Biological Science & Technology, National Yang Ming Chiao Tung University, Taipei, Taiwan; [5]Department and Graduate Institute of Pharmacology, National Taiwan University College of Medicine, Taipei, Taiwan

*For correspondence:
ben.s.lai@ibms.sinica.edu.tw

**Competing interest:** The authors declare that no competing interests exist.

**Abstract** Zebrafish exhibit a robust ability to regenerate their hearts following injury, and the immune system plays a key role in this process. We previously showed that delaying macrophage recruitment by clodronate liposome (−1d_CL, macrophage-delayed model) impairs neutrophil resolution and heart regeneration, even when the infiltrating macrophage number was restored within the first week post injury (Lai et al., 2017). It is thus intriguing to learn the regenerative macrophage property by comparing these late macrophages vs. control macrophages during cardiac repair. Here, we further investigate the mechanistic insights of heart regeneration by comparing the non-regenerative macrophage-delayed model with regenerative controls. Temporal RNAseq analyses revealed that −1d_CL treatment led to disrupted inflammatory resolution, reactive oxygen species homeostasis, and energy metabolism during cardiac repair. Comparative single-cell RNAseq profiling of inflammatory cells from regenerative vs. non-regenerative hearts further identified heterogeneous macrophages and neutrophils, showing alternative activation and cellular crosstalk leading to neutrophil retention and chronic inflammation. Among macrophages, two residential subpopulations (*hbaa*+ Mac and *timp4.3*+ Mac 3) were enriched only in regenerative hearts and barely recovered after +1d_CL treatment. To deplete the resident macrophage without delaying the circulating macrophage recruitment, we established the resident macrophage-deficient model by administrating CL earlier at 8 d (−8d_CL) before cryoinjury. Strikingly, resident macrophage-deficient zebrafish still exhibited defects in revascularization, cardiomyocyte survival, debris clearance, and extracellular matrix remodeling/scar resolution without functional compensation from the circulating/monocyte-derived macrophages. Our results characterized the diverse function and interaction between inflammatory cells and identified unique resident macrophages prerequisite for zebrafish heart regeneration.

## Editor's evaluation

The authors analyze changes in the gene expression of different immune cells during heart regeneration using single-cell RNA-sequencing and assess changes upon drug treatment that depletes macrophages. They find that drug treatment affects the gene expression profiles and abundance of different immune cells. The work provides a useful resource of gene expression data and a nice analysis supporting immune cell interactions during heart regeneration.

## Introduction

Heart failure is a major cause of morbidity and mortality, in part due to the inability of the human heart to replenish lost cardiomyocytes following myocardial infarction (MI). Unlike adult mice and humans, many vertebrates, including certain fish and amphibians, are capable of endogenous heart regeneration throughout life. As an example, zebrafish (*Danio rerio*) display a distinct ability to regenerate their heart following injury. However, this ability is not shared by another teleost, the medaka (*Oryzias latipes*) (*Lai et al., 2017*; *Ito et al., 2014*). Even mammals can regenerate their heart during embryonic and neonatal stages, despite this capacity being quickly lost postnatally (*Porrello et al., 2011*; *Haubner et al., 2016*). Comparative studies between neonatal and adult mice (*Lavine et al., 2014*; *Aurora et al., 2014*), and between phylogenetically related species such as zebrafish and medaka *Lai et al., 2017*; *Ito et al., 2014*, have suggested that the capacity for regeneration does not solely rely on genetic makeup, environmental conditions (e.g., hypoxia), or evolutionary complexity; instead, the type and extent of the immune responses to cardiac injury seem to be a major difference between these regenerative and non-regenerative models (*Lai et al., 2017*; *Lai et al., 2019*), and may largely influence the recovery post-experimental MI, as well as clinical prognosis (*Cheng et al., 2017*).

In our earlier study of reciprocal analyses in zebrafish and medaka, we observed delayed and reduced macrophage recruitment in medaka compared to zebrafish following cardiac injury. Furthermore, delaying macrophage recruitment in zebrafish by intraperitoneal (IP) injection of clodronate liposome (CL) 1 d prior to cryoinjury compromised neovascularization, neutrophil clearance, cardiomyocyte proliferation, and scar resolution, even though the number of infiltrating macrophages recovered to the control levels in the first week post injury (–1d_CL, macrophage-delayed model hereafter). These previous results indicate that late macrophages in –1d_CL-treated zebrafish were different in their identity and regenerative potential, presenting a great opportunity to further compare and learn the molecular properties of regeneration-associated macrophages. Recent studies in zebrafish also identified novel macrophage subpopulations by gene-specific reporters and their functions in cardiac repair and regeneration, hinting at a heterogeneous spectrum of cardiac macrophages (*de Preux Charles et al., 2016*; *Bevan et al., 2020*; *Sanz-Morejón et al., 2019*; *Ma et al., 2021*; *Hu et al., 2022*). However, beyond these observations, the overall heterogeneity and function of these inflammatory cells during cardiac regeneration remain unclear.

In the present study, we perform comparative bulk and single-cell transcriptomic profiling to investigate the global influence of macrophage pre-depletion and comprehensively analyze the heterogeneity, dynamic, and function of both macrophages and neutrophils in regenerative zebrafish hearts (–1d_PBS) vs. non-regenerative macrophage-delayed hearts (–1d_CL). Bulk RNAseq analysis indicated prolonged and unresolved inflammatory response and misregulated energy metabolism in –1d_CL-treated zebrafish until 3 wk post-cardiac injury, while cardiomyocyte replenishment and scar resolution took place extensively in regenerative PBS-treated control hearts. Single-cell analyses further revealed diverse macrophage subpopulations with potential functions in phagocytosis, neutrophil recruitment, reactive oxygen species (ROS) homeostasis, angiogenesis, extracellular matrix (ECM) remodeling, and inflammatory regulation during the first week post-cardiac injury. Comparative analyses between regenerative and non-regenerative hearts led to the identification of unique cardiac resident macrophage subpopulations expressing *timp4.3* and *hemoglobin* genes that potentially function in ECM remodeling, inflammatory resolution, and ROS homeostasis. Pre-depleting these resident macrophages a week or even a month prior to cardiac injury significantly impaired heart regeneration without affecting macrophage recruitment from circulation, suggesting that these resident macrophages determine the regenerative capacity and cannot be replaced by circulating/monocyte-derived macrophages. Altogether, these results unravel the heterogeneity and function of inflammatory cells during cardiac repair and highlight the indispensable role of cardiac resident macrophages in zebrafish heart regeneration.

# Results

## Delayed macrophage recruitment results in prolonged expression of genes related to inflammation and disrupted energy metabolism during cardiac repair

In zebrafish, macrophage pre-depletion at 1 d prior to cardiac injury (–1d_CL) delayed macrophage recruitment and compromised heart regeneration, even though the overall macrophage numbers were restored within the first week (*Lai et al., 2017*). To investigate the global transcriptomic changes under these regenerative (PBS-treated, normal macrophage recruitment) and non-regenerative (–1d_CL-treated, delayed macrophage recruitment) conditions, we isolated zebrafish hearts at 7 and 21 days post cryoinjury (dpci), corresponding to the time when cardiomyocytes proliferate and replace the scar tissue during heart regeneration, and subjected the hearts to bulk RNAseq analyses (*Figure 1A*). We first plotted our data from 7 and 21 dpci against published data points from 6 hours post cryoinjury (hpci) to 5 dpci in a principal component analysis (PCA, *Figure 1B*). Transcriptomes of the PBS-treated samples at 7 and 21 dpci nicely fit into the trajectory between 5 dpci and untouched hearts, suggesting that transcriptomic changes in control hearts coincide with the recovery toward the naïve state (purple dots and trail, *Figure 1B*). However, the transcriptome of non-regenerative CL-treated hearts at 21 dpci (CL21d) was in proximity to that of PBS-control hearts at 7 dpci (PBS7d), suggesting a delayed transcriptomic response in the non-regenerative CL-treated hearts (green dots, *Figure 1B*).

We next performed hierarchical clustering of all differentially expressed genes (DEGs) that behave differently under regenerative PBS vs. non-regenerative –1d_CL conditions (*Figure 1C* and *Figure 1—source data 1*). These DEGs were grouped into 22 clusters with similar expression dynamics corresponding to time points and treatments. Among regeneration-associated gene clusters, we found that genes associated with ROS homeostasis and injury repair were upregulated only in PBS hearts at 7 dpci (Cluster 2, C2), including *serpine1*, *havcr2*, *tnfaip6*, and *hmox1a*. These genes may contribute to heart regeneration through ROS regulation and inflammatory modulation (*Declerck and Gils, 2013*; *Münch et al., 2017*; *Andrews et al., 2019*; *Fang et al., 2021*; *Mittal et al., 2016*; *Chiang et al., 2018*). At 21 dpci, DEGs encoding various ribosomal subunits were upregulated only in PBS-control hearts (C9–C12), suggesting active production of building blocks for replenishing lost tissues (*Figure 1C*). Interestingly, some DEGs' functions in various metabolic processes (C19 and C20) were active in uninjured hearts, downregulated at 7 dpci, and reactivated only in PBS-treated but not CL-treated hearts at 21 dpci (*Figure 1C*). This observation corresponds nicely to the metabolic switches of CMs during cardiac regeneration. CMs in adult animals adopt oxidative metabolism after differentiation and maturation to meet the high-energy demands from constant beating (*Ellen Kreipke et al., 2016*). Upon MI, mature CMs switch back to use glucose (glycolysis) instead of fatty acid (oxidative phosphorylation) as the main substrate for energy (*Doenst et al., 2013*; *Zuurbier et al., 2020*). Interestingly, this metabolic switch was also observed during zebrafish regeneration when pyruvate metabolism and glycolysis are beneficial for CM dedifferentiation and proliferation (*Fukuda et al., 2020*).

On the other hand, the majority of the DEGs associated with non-regenerative –1d_CL condition are involved in immune-related processes, including damage-associated patterns, inflammatory cytokines, phagocytosis, and apoptotic cell death (C3–C6). These DEGs were generally expressed at 7 dpci in both PBS and CL-treated hearts, but their activation was prolonged and even intensified in CL-treated hearts at 21 dpci (*Figure 1C*). Prolonged inflammation may prevent tissue repair and regeneration (*Halade and Lee, 2022*). Inflammation also induces continuous neutrophil migration and infiltration and prevents their clearance by apoptosis (*Blume et al., 2012*). Especially in C5 and C6, DEGs involved in 'Immune response,' 'Cytokine–cytokine receptor interaction,' and 'Apoptosis' were overexpressed at 21 dpci only in CL-treated hearts, suggesting that cardiac cells (including CMs) continuously undergo programmed cell death potentially suffering from ROS stress (ferroptosis) and inflammatory microenvironment (pyroptosis) (*Olivetti et al., 1997*; *Wencker et al., 2003*). Continuous cell death subsequently triggers the prolonged activation of C-type lectin receptor singling pathway in active phagocytes (*Takeuchi and Akira, 2010*; *Campisi et al., 2014*). As a result, proteolysis of engulfed debris and cell adhesion molecules were activated under the same conditions (*Figure 1C*). Altogether, the results from hierarchical clustering of DEGs and associated GO analysis suggest that delayed macrophage recruitment disrupts inflammatory resolution, ROS homeostasis, and energy

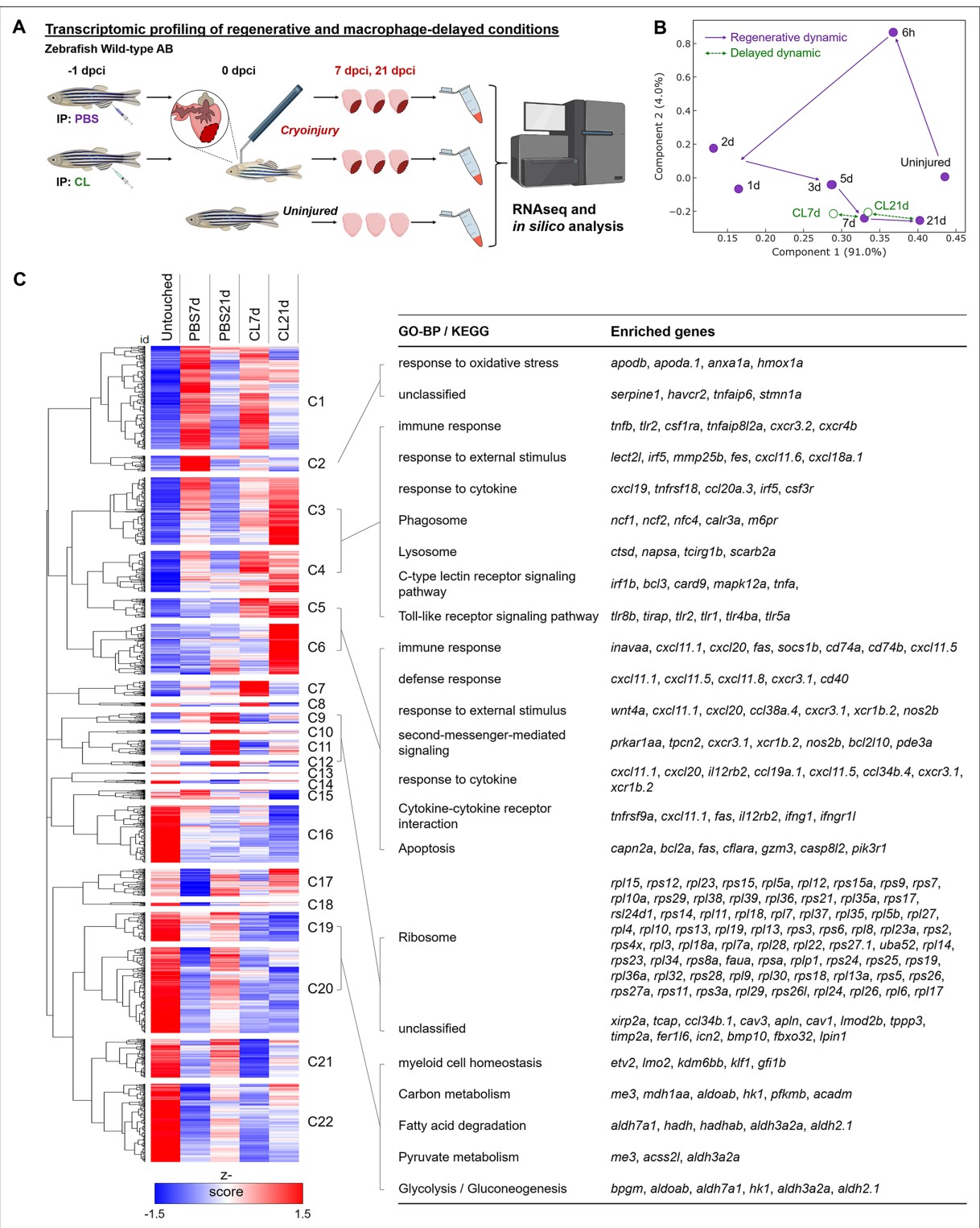

**Figure 1.** Transcriptional profiling of infarcted hearts under regenerative and macrophage-delayed conditions. (**A**) Experimental design. Zebrafish AB_wildtype was IP-injected with PBS or clodronate liposomes (CL) 1 d before cardiac cryoinjury. Injured hearts were collected at 7 and 21 days post cryoinjury (dpci), respectively. Uninjured hearts were collected as the control of the baseline. Total RNA was extracted and analyzed by RNA sequencing (RNAseq). (**B**) Principal component analysis (PCA) of gene expression in hearts at different time points. The PCA was performed on the FPKM normalized datasets of healthy hearts and injured hearts at 7 and 21 dpci after PBS or CL treatments (PBS7d, PBS21d, CL7d, and CL21d). The datasets of 0 hr, 6 hr, 1d, 2d, 3d, and 5 dpci from the previous study were also included (*Lai et al., 2017*). Regeneration and delayed dynamics are indicated by purple and green lines, respectively. FPKM, Fragments Per Kilobase of transcript per Million mapped reads. (**C**) Hierarchical clustering heatmap of the comparatively

*Figure 1 continued on next page*

*Figure 1 continued*

DEGs under regenerative and macrophage-delayed conditions in zebrafish. The DEGs were selected by NOIseq (q > 0.99) and arranged by hierarchical clustering from cluster 1 (C1) to cluster 22 (C22) (left panel). The value was a z-score from 1.5 as red to –1.5 as blue. BP of GO and KEGG pathways of the DEGs were analyzed by using WebGestalt (right panel). Cluster-enriched genes involved in their predicted biological processes and pathways were listed. The threshold of enriched categories was FDR < 0.05. DEGs, differentially expressed genes; GO, Gene Ontology; BP, biological process; KEGG, Kyoto Encyclopedia of Genes and Genomes; FDR, false discovery rate.

The online version of this article includes the following source data and figure supplement(s) for figure 1:

**Source data 1.** Full DEGs list from RNAseq data of -1d-CL vs. PBS-control hearts at 7 and 21 days post cryoinjury.

**Figure supplement 1.** Identification of canonical pathways and upstream regulators under regenerative and macrophage-delayed conditions.

metabolism during cardiac repair. The full list of DEGs in each cluster is summarized in *Figure 1— source data 1*.

To identify the canonical pathways and potential upstream regulators associated with aberrant regeneration, these DEGs were further analyzed by Ingenuity Pathway Analysis (IPA) (*Figure 1—figure supplement 1*). –1d_CL treatment led to continuous activation of 'Leukocyte Extravasation Signaling' and 'Production of NO and ROS in Macrophages' pathways (*Figure 1—figure supplement 1A*). Correspondingly, downstream genes of inflammatory cytokines IFNG, TNF, and IL6 were continuously activated in CL-treated hearts, while these pathways were largely downregulated in PBS-control hearts at 21 dpci (*Figure 1—figure supplement 1B*). Among genes of the 'Leukocyte Extravasation Signaling' pathway, we found several integrin genes usually expressed on the leukocyte plasma membrane, including *itga4*, *itgal,* and *itgb2*, which are involved in leukocyte enrolling on endothelial cells and transmigration (*Mitroulis et al., 2015*; *Herter and Zarbock, 2013*). In addition, we found several matrix metalloproteinases (MMPs), including *mmp9, mmp13,* and *mmp25*, which might be involved in ECM remodeling and leukocyte recruitment during inflammation (*Figure 1—figure supplement 1C*; *Song et al., 2013*; *Bradley et al., 2012*; *Starr et al., 2012*). These results support our previous observation of continuous neutrophil infiltration and retention in the CL-treated hearts even until 30 dpci (*Lai et al., 2017*). Lastly, among genes of the 'NO and ROS production in Macrophages' pathway, we found continuous activation of DAMP/PAMP receptor *tlr2*, neutrophil cytosolic factors (*ncf1*, *ncf2*, and *ncf4*), myeloid cell-lineage committed gene *spi1*, and its downstream target *ptpn6*, in addition to the macrophage differentiation marker *irf8* in CL-treated hearts at 21 dpci (*Figure 1—figure supplement 1D*). Collectively, these results indicate that macrophages may play roles in regulating ROS homeostasis, immune cell dynamics, and inflammation resolution as these processes were misregulated under –1d_CL treatments and associated with impaired heart regeneration.

## Single-cell analyses reveal the heterogeneous landscape and dynamic changes of inflammatory cells during cardiac repair

Since macrophage properties may be altered upon –1d_CL treatment and thus fail to support heart regeneration by resolving neutrophil infiltration and inflammation, we analyzed and compared the potential identity and function of these inflammatory cells by single-cell transcriptomic profiling (*Figure 2A*). Adopting the same macrophage-delayed model (–1d_CL treatment), double transgenic zebrafish *Tg(mpx:EGFP;mpeg1:mCherry)* were IP injected with CL or PBS at 1 d before cryoinjury, and the EGFP-expressing neutrophils and mCherry-expressing macrophages were isolated by fluorescence-activated cell sorting (FACS) from uninjured hearts, as well as regenerative PBS-control and non-regenerative –1d_CL treated hearts at 1, 3, and 7 dpci (*Figure 2A* and *Figure 2—figure supplement 1A*; *Bernut et al., 2014*; *Mathias et al., 2009*). In uninjured hearts, we found a substantial number of mCherry$^+$ and mCherry$^+$EGFP$^+$ resident cells (mostly macrophages, ~0.6% of total cardiac cells) and very few EGFP$^+$ neutrophils (*Figure 2—figure supplement 1A*). Among them, mCherry$^+$EGFP$^+$ cells show higher complexity and larger size (FCS-A and SSC-A) than mCherry$^+$ cells, corresponding to the macrophages and the progenitor/lymphocyte properties previously described (uninjured sample in *Figure 2—figure supplement 1A*; *Traver et al., 2003*). After injury, both macrophages and neutrophils increased rapidly, and divergent cell composition and numbers were observed in PBS vs. CL-treated hearts over time. While macrophage numbers increased after cardiac injury until 7 dpci, neutrophils peaked at 3 dpci and gradually decreased between 3 and 7 dpci in PBS-control hearts, corresponding to the inflammatory resolution phase previously described (*Figure 2—figure*

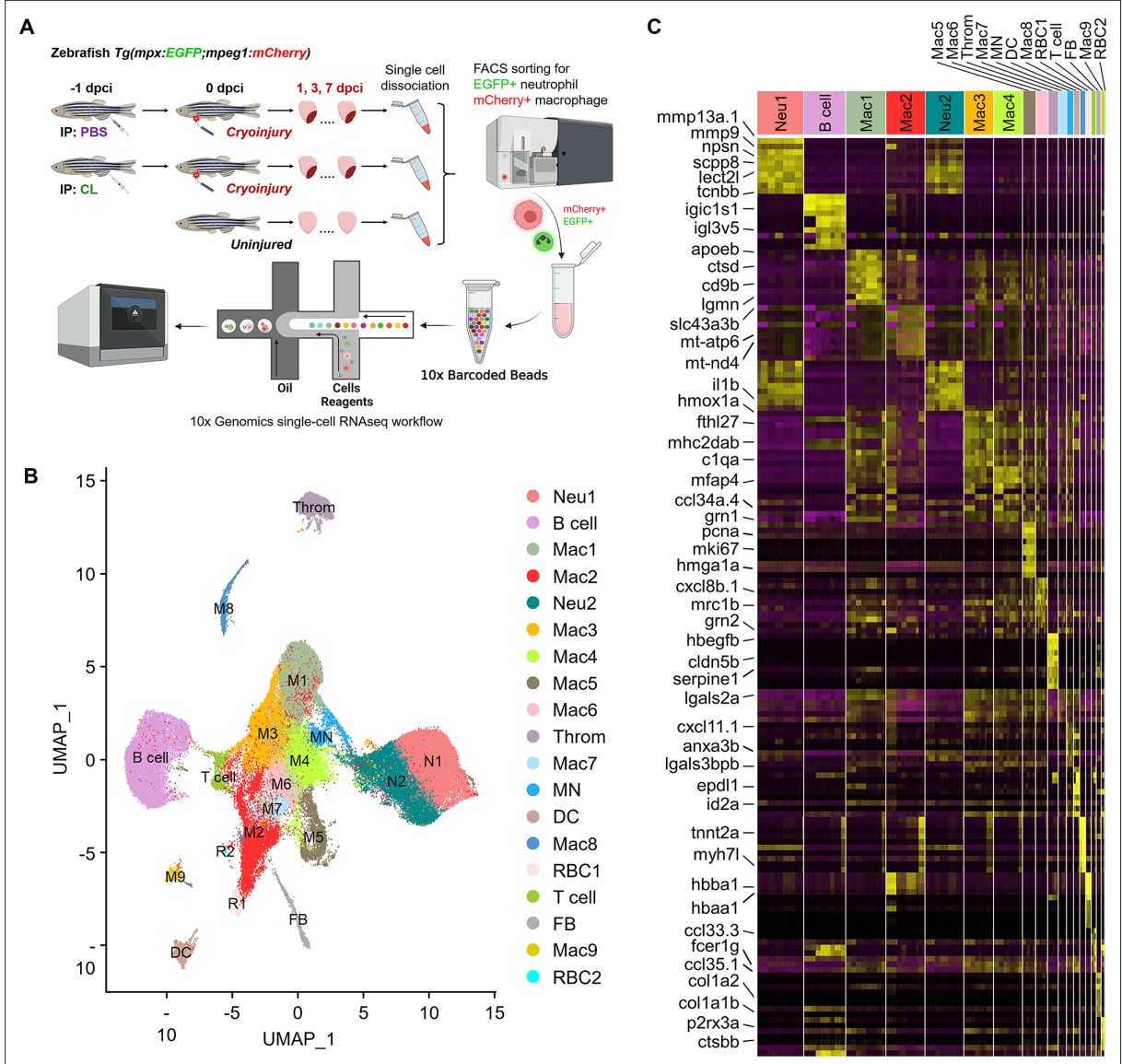

**Figure 2.** Temporal single-cell analyses revealed heterogeneous macrophages and neutrophils in the infarcted hearts. (**A**) Experimental design. Double transgenic *Tg(mpx:EGFP;mpeg1:mCherry)* zebrafish expressing EGFP in neutrophils and mCherry in macrophages were IP-injected with PBS (regenerative condition) or CL (macrophage-delayed condition) 1 d before cryoinjury (–1 dpci, –1d_CL). Injured hearts were collected and dissociated into single cells at 1, 3, and 7 days post cryoinjury (dpci). Untreated and uninjured hearts were also collected and dissociated. Single cells of each time point were then subjected to a fluorescence-activated cell sorter (FACS) for isolating the mCherry⁺ and EGFP⁺ cells. RNA was purified from these cells and barcoded followed by single-cell RNA sequencing (scRNAseq). (**B**) Uniform Manifold Approximation and Projection (UMAP) of the isolated cells. The isolated cells consisted of nine macrophage clusters, two neutrophil clusters, one hybrid cluster (MN), and other minor populations including B cell, thrombocyte (Throm), dendritic cell (DC), T cell, fibroblast (FB), and two red blood cell (RBC) clusters. (**C**) Heatmap of top 10 DEGs in 19 clusters of infarcted hearts. Yellow highlights the cluster-enriched genes with gene names listed on the left. DEGs, differentially expressed genes.

The online version of this article includes the following source data and figure supplement(s) for figure 2:

**Source data 1.** R script of cell cycle regression analysis of scRNAseq data.

**Source data 2.** Cluster-enriched genes of each cell cluster from scRNAseq analysis.

**Figure supplement 1.** Dynamics of macrophages and neutrophils in zebrafish heart prior and post injury.

**Figure supplement 2.** Quality controls for all the scRNAseq datasets by Seurat.

**Figure supplement 3.** Expression of marker genes visualized on Uniform Manifold Approximation and Projection (UMAP) plots.

**Figure supplement 4.** Cell-cycle scoring and regression of cell clusters revealed by scRNAseq.

*supplement 1A*; *Bevan et al., 2020*). In CL-treated hearts, macrophages progressively increased, similar to control hearts, but neutrophil numbers became much higher than controls at both 3 and 7 dpci (*Figure 2—figure supplement 1A*). At 7 dpci, a similar percentage of macrophages were sorted under both conditions, while a higher percentage of neutrophils were sorted in CL conditions (*Figure 2—figure supplement 1B*) in line with our previous findings that neutrophil resolution was delayed in CL-treated hearts (*Lai et al., 2017*). Interestingly, EGFP⁺/mCherry⁺ macrophages resided nearby the epicardial layer of the uninjured/naïve hearts and proliferated to maintain their population, similar to murine resident macrophages (*Figure 2—figure supplement 1C and D*; *Epelman et al., 2014*). Upon cardiac injury, those resident macrophages were preferentially enriched in regenerative hearts at 1–3 dpci (*Figure 2—figure supplement 1A*) and populated the infarct area at 7 dpci (*Figure 2—figure supplement 1E*).

Inflammatory cells were subjected to droplet-based high-throughput single-cell RNA sequencing (scRNAseq). To visualize the dataset, sequencing reads were mapped to the zebrafish genome, assigned to each cell, and then processed by the Uniform Manifold Approximation and Projection (UMAP) for dimension reduction and unbiased clustering using Seurat R package (*Figure 2B* and *Figure 2—figure supplement 2*; *Butler et al., 2018*). After clustering, we identified 19 distinct clusters of inflammatory cells and found that all of them expressed myeloid lineage marker genes *spi1b* and *coro1a* (*Figure 2B* and *Figure 2—figure supplement 3A*). Among them, nine macrophage clusters (Cluster Macs), two neutrophil clusters (Cluster Neus), and one hybrid cluster (Cluster MN) were identified, based on the expression of reporter genes *mpeg1* and *mpx*, as well as other mononuclear phagocyte markers *csf1r*, *ccr2*, *cxcr1*, *irf8*, *lyz*, *mfap4*, and *kita* (*Figure 2B* and *Figure 2—figure supplement 3A*). We also identified small populations of B cells, T cells, dendritic cells, thrombocytes, red blood cells, and fibroblasts based on respective marker genes shown in UMAP and heatmap (*Figure 2B* and *Figure 2—figure supplement 3B*). Correspondingly, a subpopulation of B cells has been previously shown to express *mpeg1* and observed in *mpeg1:mCherry* fish (*Ferrero et al., 2020*). Minor *mpeg1⁻/mpx⁻* clusters might come from contamination during FACS, even though stringent gating strategies were applied, and will not be further analyzed in this study (*Farbehi et al., 2019*; *Dick et al., 2019*).

Besides common lineage markers, heterogeneous macrophage subsets exhibited cluster-enriched/specific gene expression (*Figure 2C*). Unlike macrophages, neutrophils were classified into only two populations Neu 1 and Neu 2, which both seem to be mature and express granular genes and integrins (*Figure 2C* and *Figure 2—figure supplement 3C*; *Xie et al., 2020*). Notably, interferon-stimulated genes such as *rsad2*, *isg15*, *ifit8,* and *mxa* were enriched in the Neu 2 and the *mpeg1⁺* macrophages MN (*Figure 2—figure supplement 3D*). These results revealed the heterogeneous landscape of inflammatory cell subpopulations in the zebrafish injured hearts and the markers may be used to differentiate respective cluster and generate new tools for functional analysis in the future. Notably, the diverse clusters remain the same regardless of their cell-cycle-related properties/genes expression (*Figure 2—figure supplement 4*). The full list of cluster-enriched genes is summarized in *Figure 2—source data 2*.

## Temporal cell proportion analyses identified specific resident macrophage subsets associated with heart regeneration

To dissect the dynamic changes of these inflammatory cell clusters under regenerative PBS vs. non-regenerative –1d_CL conditions, we generated split UMAPs for each time point and condition (*Figure 3A*). In uninjured heart/naïve state, Mac 2 and 3 represent the major resident macrophage clusters followed by Mac 1, 4, and 8 (*Figure 3B*). The proliferating macrophage cluster Mac 5 was also observed in the uninjured hearts, corresponding to those residing nearby the epicardial tissue, suggesting that some resident macrophages may self-renew through local proliferation (*Figure 2—figure supplement 1C*; *Ma et al., 2018*). In the regenerative PBS-control hearts, Mac 1, 4, and 5 increased quickly at 1 dpci and gradually reduced back to a steady state at 7 dpci, while Mac 2 and 3 expanded substantially over the first week post injury (*Figure 3B*). On the contrary, we noticed a dramatic reduction of Mac 2 and retention of Mac 1, 4, 5, and 6 in the –1d_CL-treated hearts over time after injury (*Figure 3B*). Lastly, the minor resident populations Mac 8 and 9 were diminished after cardiac injury and barely recovered in both conditions (*Figure 3B*). We then calculated the cluster contribution toward regenerative vs. non-regenerative conditions in a cell proportion analysis

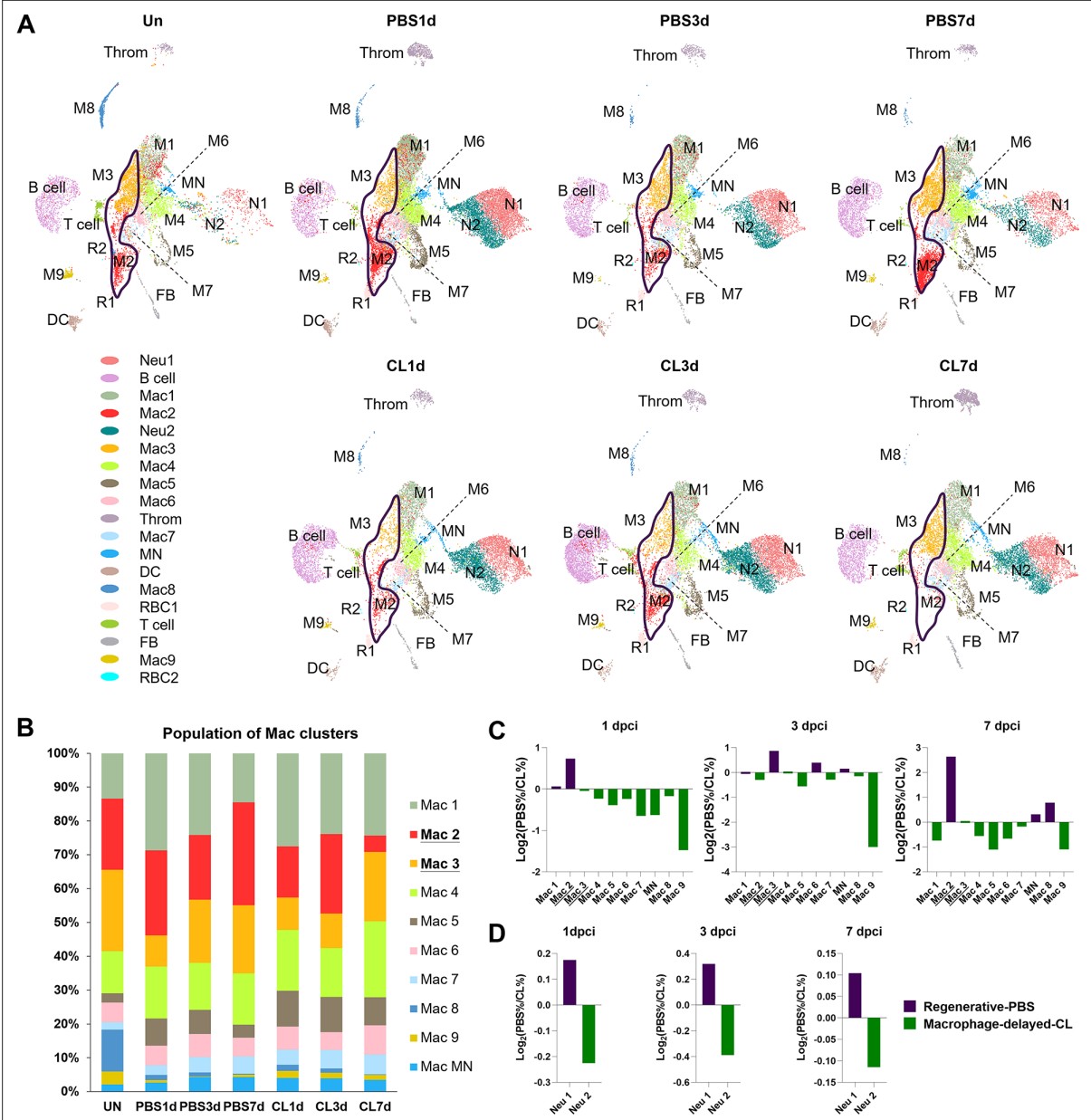

**Figure 3.** Temporal cell proportion analyses of inflammatory cells identified resident macrophage clusters enriched in regenerative conditions. Differential proportion analyses of macrophage and neutrophil clusters under regenerative or macrophage-delayed conditions. (**A**) Split view of Uniform Manifold Approximation and Projection (UMAP) plots of major macrophage (Mac) and neutrophil (Neu) clusters as well as minor inflammatory cell clusters from uninjured (UN) and infarcted hearts under regenerative (PBS) or macrophage-delayed (CL) conditions. Mac 2 and Mac 3 clusters were the major resident macrophages enriched in regenerative conditions (delineated by purple lines), and they either dramatically decreased or barely recovered in non-regenerative conditions. (**B**) The stacked bar chart shows the percentage of macrophage clusters at each time point and condition. (**C, D**) Cell proportion analyses identified the regenerative-associated clusters (purple) and macrophage-delayed-associated clusters (green) of macrophages (**C**) and neutrophils (**D**). Proportion of each cell clusters under regenerative conditions vs. macrophage-delayed conditions are shown by log2 ratio.

The online version of this article includes the following source data and figure supplement(s) for figure 3:

**Source data 1.** RNAseq data of injured ventricles of PBS vs. -1d_CL treated zebrafish at 7 and 21 days post cryoinjury.

**Figure supplement 1.** Representative genes expressed in the major resident macrophage clusters Mac 2 and Mac 3 of infarcted heart.

(*Figure 3C and D*). Coincidently, macrophage clusters enriched in regenerative conditions are the major resident clusters Mac 2 and 3, which were either dramatically decreased or barely recovered in CL-treated hearts (*Figure 3C*). To examine the key enriched genes of major resident clusters Mac 2 and 3, the hemoglobin genes *hbaa1* and *timp4.3* were validated by in situ hybridization (ISH) in injured hearts at 7 dpci (*Figure 3—figure supplement 1A and B*). Indeed, we observed Mac 2-enriched gene *hbaa1* expression specifically in RBCs (cells with olive shape labeled by asterisks in *Figure 3—figure supplement 1B*) and macrophage-like cells (cells labeled by arrows in *Figure 3—figure supplement 1B*). We also observed the Mac 3-enriched gene *timp4.3* expression in macrophage-like cells in the injured area and the epicardial layer (cells labeled by arrows in *Figure 3—figure supplement 1B*). Low *ccr2* expression in these resident macrophages suggests that they might originate from embryonic-derived lineage instead of circulatory/monocyte-derived lineage (e.g., Mac 1, *Figure 2—figure supplement 3A* and *Figure 3—figure supplement 1C*; *Lavine et al., 2014*; *Epelman et al., 2014*).

On the other hand, Neu 1 and Neu 2 were two heterogeneous neutrophil clusters actively recruited to hearts after cardiac injury. We observed a decrease of both clusters from 3 to 7 dpci in the PBS-control hearts, while they retained in the CL-treated hearts by 7 dpci (*Figure 3A*). The proportion of Neu 1 was slightly higher than Neu 2 under regenerative conditions throughout the first week post cardiac injury (*Figure 3D*). These results delineate the dynamic changes of each inflammatory cell cluster and identified regeneration-associated resident macrophages preferentially enriched in PBS-control hearts, which might play pivotal roles in cardiac regeneration.

## Alternative activation of inflammatory cells during cardiac repair under regenerative vs. macrophage-delayed conditions

To investigate the potential function of these heterogeneous inflammatory cells, we extracted and analyzed the overall DEGs of each cluster (*Table 1*, cluster-enriched genes; full list is given in *Figure 2—source data 2*) and their differential changes toward regenerative conditions (PBS-enriched) vs. non-regenerative conditions (CL-enriched) (*Figure 4*, condition-enriched genes, full list is given in *Figure 4—source data 1*). We aimed to identify cluster-enriched/specific markers of each macrophage subpopulation, and their function associated with regenerative vs. fibrotic repair by condition-enriched DEGs. Both macrophage origins and functional polarization may influence their gene expression profiles, so we perform Gene Ontology (GO) analyses based on each cell clusters and their DEGs in regenerative vs. macrophage-delayed conditions to further depict their functions during cardiac repair and regeneration (*Figure 4*).

Strikingly, most macrophage and neutrophils clusters exhibited alternative activation of different sets of DEGs in PBS vs. CL-treated conditions, except for those resident macrophage Mac 2, 3, and 8, which mostly show DEGs in regenerative PBS condition (*Figure 4*). Mac 2 exhibited high expression levels of hemoglobin genes, including *hbba1* and *hbaa1*, which are expressed in murine macrophages stimulated with LPS or interferon gamma and may be involved in NO signaling, as well as *romo1*, *prdx2*, and *hmox1a*, which are involved in the regulation of ROS (*Figure 4A*, D2; *Liu et al., 1999*; *Straub et al., 2012*; *Pérez-Torres et al., 2020*; *Chung et al., 2006*; *Yang et al., 2007*). *hmox1a* specifically functions in heme degradation, iron homeostasis, and inflammatory modulation (*Vijayan et al., 2018*; *Tomczyk et al., 2019*). These results suggest that Mac 2 may involve in homeostasis of NO, ROS, and heme during the inflammatory and resolution phase, while they are diminished in CL-treated hearts at 7 dpci (*Figure 3A and B*). Mac 3 preferentially expressed cardiac protective gene *cd74* (receptor for macrophage migration inhibitory factor) (*Rassaf et al., 2014*), myeloid cell lineage marker *spi1a* (*Bennett et al., 2001*), and metallopeptidase inhibitor *timp4.3* (*Matchett et al., 2019*; *Koskivirta et al., 2010*), suggesting a more progenitor-like status and functions related to immune modulation and ECM remodeling (*Table 1* and *Figure 2—source data 2*). Mac 8 preferentially expressed CM structural genes such as *tnnt2a* and *myh7l*, and other genes involved in muscle structure and heart development, similar to the previously reported $CX_3CR_1^+$ cardiac resident macrophages in mice (*Figure 2—source data 2*; *Walter et al., 2018*).

Among macrophage clusters exhibiting alternative activation between regenerative PBS and non-regenerative –1d_CL conditions, Mac 1 displayed the most diverse gene expression/functions in response to cardiac injury (*Figure 4*). Under regenerative condition, Mac 1 expressed tissue repairing genes related to angiogenesis, cardiovascular system development, debris clearance, and ECM composition, including *vegfaa/bb*, *lrp1ab*, *elmo1*, and *fn1a* (*Figure 4*, D11; *Marín-Juez et al.,*

**Table 1.** Biological process (BP) of Gene Ontology (GO) and Kyoto Encyclopedia of Genes and Genomes (KEGG) pathways for differentially enriched genes in each inflammatory cell clusters.

| Cluster | GO-BP term | Enriched genes | KEGG | Enriched genes |
|---|---|---|---|---|
| Mac 1 | Vesicle-mediated transport cation transport | havcr1, marco, igf2r | Endocytosis | igf2r |
| | Proton transmembrane transporter | | Ferroptosis | Heme metabolism (hmox1a) |
| | | (atp6v0ca, atp6v1e1b) | | |
| | | Solute carrier family (slc2a6, slc30a1a) | | |
| Mac 2 | Oxygen transport | Hemoglobins (hb genes), myoglobin (mb) | Unclassified | Heme metabolism (hmox1a) |
| | Hydrogen peroxide metabolic process | Hemoglobins (hb genes) | | |
| Mac 3 | Immune system process | cd74a, csf1ra | Ferroptosis | Heme metabolism (hmox1a) |
| | regulation of cell differentiation | lgals2a, spi1a | Apoptosis | Pro-apoptotic gene (pmaip1) |
| | Unclassified | timp4.3 | | |
| Mac 4 | Response to oxidative stress | prdx2, anxa1a | Citrate cycle (TCA cycle) | TCA cycle-related genes |
| | | | | (suclg1, suclg2, mdh1aa, sdha, dlst) |
| | Response to wounding | cxcl8a, lgals2a | | |
| Mac 5 | Mitotic cell cycle | cdk1, top2a | DNA replication | pcna |
| Mac 6, 7 | Response to oxidative stress | anxa1a, prdx2, park7, hmox1a | Cardiac muscle contraction | Cytochrome c oxidase involved |
| | | | | in oxidative phosphorylation |
| | Immune system development | Neutrophil transmigration (anxa1a, cx43) | Ribosome | |
| | | Leukocyte differentiation (ak2) | | |
| Mac 8 | Mitochondrion organization | NADH dehydrogenase (ndufs1, ndufs8a) | Cardiac muscle contraction | tnnt2a, myh7l |
| | | Ubiquinol-cytochrome c reductase | | |
| | | (uqcr10, uqcrc2b) | | |
| | Mitochondrial transport | vdac2, uqcrc2a, cyc1 | Glycolysis/gluconeogenesis | aldoab, pdha1a |
| | Muscle structure development | pgam2, desma, csrp3 | Fatty acid degradation | acadvl, acsl1b |
| | Heart development | fabp3, myl7, nppa | | |
| Mac 9 | Regulation of cell differentiation | irf8, jun, myd88 | Endocytosis | cxcr4a, cxcr4b, spg21, eps15 |
| | Cellular macromolecule localization | cd74a, cd74b | Toll-like receptor signaling pathway | Toll-like receptors (tlr3, tlr8b, tlr9) |
| | | | | Interferon-induced genes (irf5, irf7) |
| MN | Immune response | Inflammation-related genes (tnfb, irak3) | Mitophagy | Autophagy-involved genes (gabarapa, gabarapb, gabarapl2, calcoco2) |
| | Response to cytokine | Cytokines and receptors | NOD-like receptor signaling pathway | Interferon-induced genes (stat1b, irf1b) |

*Table 1 continued on next page*

*Table 1 continued*

| Cluster | GO-BP term | Enriched genes | KEGG | Enriched genes |
|---------|-----------|----------------|------|----------------|
| | | (*cxcl11.1, cxcl20, tnfrsf18, ccr9a*) | | |
| | | | C-type lectin receptor signaling pathway | Fc receptor (*fcer1g*), |
| | | | | adaptor protein of PRR (*card9*) |
| | | | Ferroptosis | *hmox1a* |
| Neu 1 | Unclassified | *sat1a.2, raraa* | Ribosome | *rplp0, rpsa, rpl15* |
| Neu 2 | Immune response | *ccl34b.8, tnfb* | Phagosome | *nfc1, nfc2* |
| | antigen processing and presentation | *cd74a, cd74b* | Lysosome | *ctss2.2, ctsba, tcirg1b* |
| | Cellular response to nitrogen starvation | *map1lc3b, gabarapl2, gabarapb* | NOD-like receptor signaling pathway | *il1b*, inflammasome related gene |
| | | | | (*jun, txnipa*) |
| | atp metabolic process | *pgam1a, pkma* | Glycolysis/gluconeogenesis | *pkma, pgam1a, aldocb* |
| | | | C-type lectin receptor signaling pathway | *egr3, il1b, irf1b* |
| | | | Mitophagy | *gabarapl2, atf4a, gabarapb* |
| | | | *Salmonella* infection | *il1b, tlr5b, fosab* |

*2016*; *Pi et al., 2012*; *Fernandez-Castaneda et al., 2013*; *Epting et al., 2010*; *Wang et al., 2013*). Under non-regenerative condition, Mac 1 expressed inflammatory cytokines *il1b* and *tnfaip2b*, and genes associated with autophagy and RNA splicing (*Figure 4*, D6, D10, and D13; *Janssen et al., 2020*; *Wu and Lu, 2019*). High *ccr2* expression also suggests that Mac 1 might have originated from monocyte-derived lineage (*Figure 3—figure supplement 1C*). Among minor clusters, Mac 4, 5, 6, and 7 showed higher expression of genes related to oxidative stress under regenerative condition, including *prdx2*, *anxa1a*, and *lgals2a* (*Figure 4*, D5), suggesting that these macrophage subpopulations might have roles in ROS homeostasis and facilitating inflammation resolution (*McArthur et al., 2020*; *Solito et al., 2003*). On the contrary, these macrophage subsets expressed chemokine for neutrophil recruitment, transmigration, and differentiation (*Table 1* and *Figure 4*, D13; *de Oliveira et al., 2013*), as well as incomplete fatty acid metabolism genes under non-regenerative condition (*Figure 4*, D16 and D6, and *Figure 2—source data 2*; *Thorp, 2021*), which may lead to extended neutrophil function, inflammation, and ROS generation (*Table 1*). Corresponding to the traditional concept of M1/M2 macrophage polarization, Mac 1, 4, and 5 preferentially expressed inflammatory genes, including *il1b*, *tnfb*, and *ifngr2*, while Mac 2 and 3 preferentially expressed the M2 markers *arg2* and *mrc1b* (*Figure 4—figure supplement 1A*; *Bevan et al., 2020*; *Dick et al., 2022*; *Tsuruma et al., 2018*; *Dowling et al., 2021*; *Xu et al., 2020*). Taken together, detailed analyses on both cluster-enriched and condition-enriched genes suggest that circulation-recruited/monocyte-derived macrophages may exhibit distinct functional polarization toward continuous neutrophil recruitment and inflammation upon –1d_CL pre-depletion, reflecting the complex nature of macrophage polarization and functions in cardiac repair.

Alternative genes activation under regenerative vs. non-regenerative conditions were also observed in neutrophil clusters, especially for Neu 1 (enriched under the regenerative condition in *Figure 3D*). Neu 1 upregulated retinoic acid receptor *raraa*, necroptosis genes *fth1a*, *caspa*, and *hmgb1a*, and inflammatory modulation genes *cul1b* and *spop* under regenerative condition (*Figure 4*, D14), which corresponds nicely with the fact that programmed cell death and phagocytosis are the main mechanisms of neutrophil clearance and inflammatory resolution (*Lepilina et al., 2006*; *Kikuchi et al., 2011*; *Greenlee-Wacker, 2016*). On the other hand, both Neu 1 and Neu 2 expressed genes related to typical neutrophil functions under non-regenerative condition, including vesicle transport, phagocytosis, energy metabolisms, regulation of proteolysis, and oxidative phosphorylation (*Figure 4*, D6, and *Figure 2—source data 2*). Besides these common genes, Neu 2 further express *atp6v1g1* and

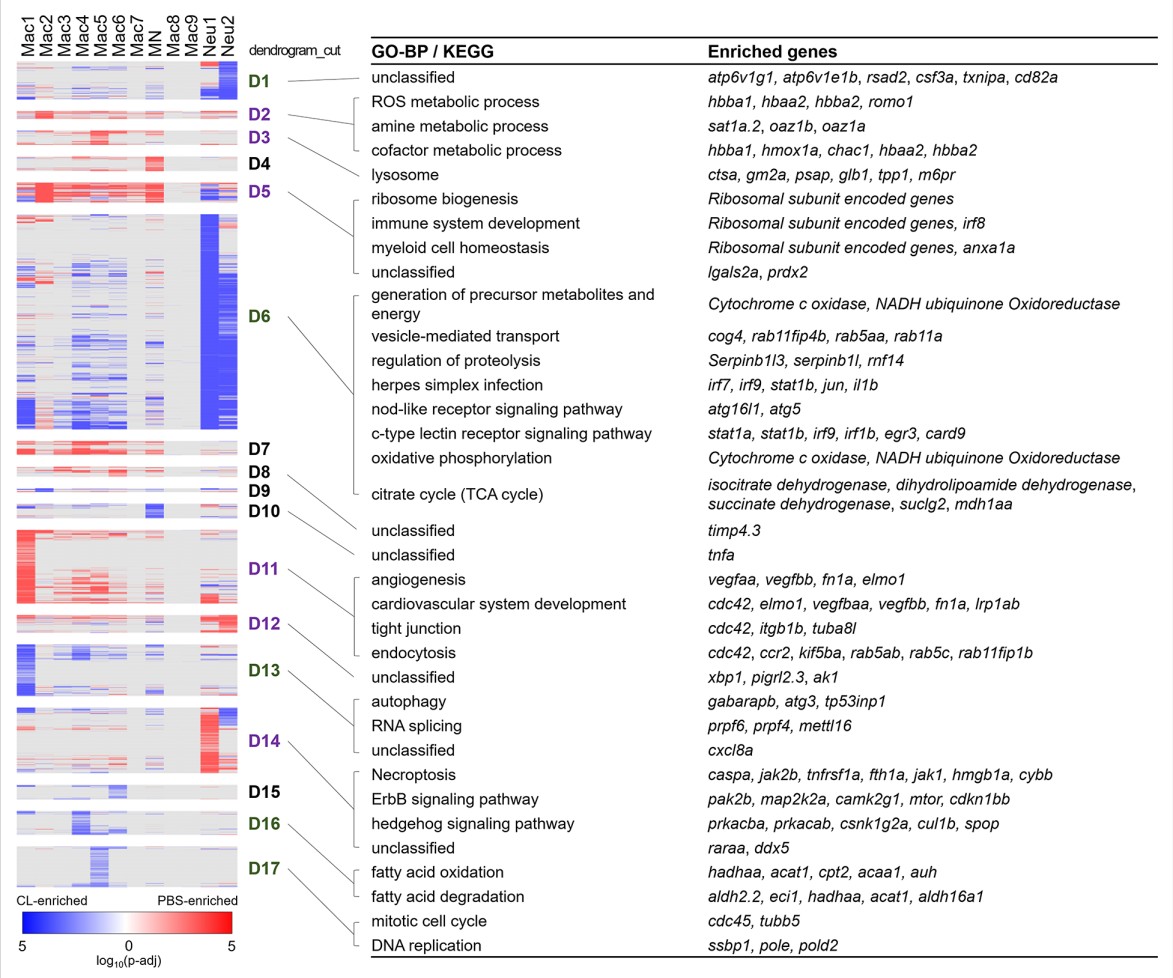

**Figure 4.** Differential gene expression in respective inflammatory cell clusters toward regenerative vs. macrophage-delayed conditions revealed alternative activation of both macrophages and neutrophils. Hierarchical clustering of the condition-enriched DEGs between PBS and CL conditions of each macrophage and neutrophil cluster. PBS-enriched and CL-enriched genes are highlighted in red and blue, respectively. D1–D17 represent the dendrogram cut of the hierarchical clustering. BP of GO and KEGG pathways were identified from the DEGs in respective dendrograms. Regenerative and macrophage-delayed associated dendrograms are labeled by purple and green, respectively. DEGs, differentially expressed genes; GO, Gene Ontology; BP, biological process; KEGG, Kyoto Encyclopedia of Genes and Genomes.

The online version of this article includes the following source data and figure supplement(s) for figure 4:

**Source data 1.** Conditional-enriched genes of each cell cluster from scRNAseq analysis.

**Figure supplement 1.** Expression pattern of classical macrophage polarization genes and the alternatively activated genes between Neu 1 and Neu 2.

*atp6v1e1b,* which encode for components of vacuolar ATPase that mediates vesicular acidification and contributes to pH-related inflammatory responses (*Pamarthy et al., 2018*) and interferon (IFN)-stimulated gene *rsad2* (*Figure 4*, D1; *Rivera-Serrano et al., 2020*). When comparing these two neutrophil clusters directly, Neu 1 preferentially expressed granule genes for ECM remodeling and cell signaling, including *lyz, npsn, mmp13a.1,* and *mmp9,* while Neu 2 expressed inflammatory cytokines *il1b* and *tnfb,* phagosome-related genes *ncf1* and *ncf2,* as well as neutrophil chemotaxis genes *cxcr1, atf3,* and *illr4* (*Table 1* and *Figure 4—figure supplement 1B*). Taken together, these results suggest that Neu 1 enriched in the regenerative (PBS-control) condition and functions in debris clearance, inflammatory modulation, and turn on programmed cell death for its own clearance. On the contrary, Neu 2 enriched in the macrophage-delayed model and functions in inflammatory propagation and recruitment of more inflammatory cells, nicely explaining the continuous neutrophil recruitment and retention that we observed in the CL-treated heart.

## Cellular crosstalk analysis indicates that resident macrophages mediate ECM remodeling and phagocytic clearance of neutrophils

Since our previous study and current findings indicate that neutrophils retain in the injury associated with unresolved inflammation when macrophage properties change in –1d_CL-treated hearts, we further investigate the cell–cell interactions between macrophages and neutrophils under regenerative and non-regenerative conditions (*Figure 5*). Neutrophils are recruited to the injured tissue by various cytokines and chemokines and programmed for cell death as soon as they clear the tissue debris together with other professional phagocytes (*Kim and Luster, 2015*). Apoptotic neutrophils are then cleared by macrophages to prevent further release of cytotoxic and inflammatory components, which is a critical step of inflammatory resolution (*Greenlee-Wacker, 2016*). These interactions are largely mediated by ligand–receptor interactions, so we first established the ligand–receptor pairs between macrophage and neutrophil clusters based on a published pipeline and then sorted them according to PBS-specific and CL-specific crosstalk at different time points post injury (*Figure 5* and *Figure 5—source data 1–4*; *Ramilowski et al., 2015*).

Under regenerative condition, resident macrophages Mac 2 and 3 seem to signal and regulate neutrophil migration at 3 dpci and reverse migration and resolution at 7 dpci (*Figure 5A and B*). For example, Mac 2 and 3 may stimulate neutrophil recruitment via complement component 3 (C3)–integrin ITGA1 (ITGAX/ITGB2) and afadin (AFDN)–junctional adhesion molecule-A (JAM-A) interaction (*Houseright et al., 2020*; *Woodfin et al., 2009*; *Vandendriessche et al., 2021*; *Wang and Liu, 2022*) and regulation cholesterol homeostasis and inflammasome in neutrophils through PLTP–ABCA1 axis at 3 dpci (*Figure 5A′*; *Westerterp et al., 2018*; *Jiang et al., 2012*). At 7 dpci, adhesion, survival, and migration of neutrophils could be influenced by resident macrophages via ECM remodeling through ADAMs, COL1A1, FN1, and TIMP2 expression (*Figure 5B′*; *Deligne and Midwood, 2021*). Particularly, HMGB1 signaling to CXCR4 is critical for neutrophil reverse migration, while neutrophil retention was observed when CXCR4 is blockaded in sterile inflammation (*Wang et al., 2017*). The result suggested that neutrophils were in high motility and left the injured site by reverse migration in the PBS-control hearts, which were missing in the macrophage-delayed hearts. In addition, promotion of neutrophil self-phagocytosis could be mediated by the MFGE8–ITGAV axis under regenerative condition at 3 and 7 dpci (*Figure 5A′ and B′*; *Hanayama et al., 2002*; *Ravichandran, 2010*; *Siakaeva et al., 2019*).

Under non-regenerative condition, resident macrophages seem to modulate ECM earlier at 3 dpci via the similar molecules ADAMs, COL1A1, F13A1, FN1, TIMP2, LAMB1, and TGFB1 and propagate inflammation and neutrophil survival via cytokines IL15, IL16, and TNF (*Figure 5A″*; *Mathy et al., 2000*; *Richmond et al., 2014*). These ligand–receptor pairs indicate differential neutrophil behavior resulted from altered macrophage properties and function, especially regarding the dynamic change in ECM remodeling, leading to enhanced neutrophil recruitment and/or retention under non-regenerative condition.

On the other hand, neutrophil–ligands to macrophages–receptors pair also showed dramatic differences between regenerative and non-regenerative conditions (*Figure 5C and D*). Under regeneration condition, the ligand–receptor pairs were mainly involved in phagocytic clearance when neutrophils express multiple eat-me/find-me signals recognized by macrophage receptors LRP1 and INTEGRINS, leading to neutrophil resolution. For example, neutrophils express calreticulin (CALR), a well-known 'eat-me' signal, recognized by phagocytic receptors- LRP1 and SCARF1 on Mac 4, 5, 6, and 8 (*Figure 5D′*; *Ravichandran, 2010*; *Gardai et al., 2005*; *Ramirez-Ortiz et al., 2013*). Furthermore, we observed various interactions mediated by the fas-associated death domain (FADD), FADD–TRADD, and FADD–ABCA1 axes, which are related to the initiation of neutrophil apoptosis (*Figure 5D′*; *Sun et al., 2000*; *Croker et al., 2011*). In contrast, neutrophils expressed multiple ligands triggering NOTCH2 signaling that correlates with proinflammatory M1 macrophage polarization and the murine Ly6C[hi] monocyte differentiation under non-regenerative condition (*Figure 5D″*; *Xu et al., 2015*; *Gamrekelashvili et al., 2020*). Taken together, these results indicate that neutrophils avoid programed cell death and phagocytic clearance by macrophages under non-regenerative condition, leading to their retention in hearts with propagated inflammation. Retained neutrophils may in turn exacerbate M1/proinflammatory macrophage polarization instead of pro-resolving/tissue-repairing phenotypes.

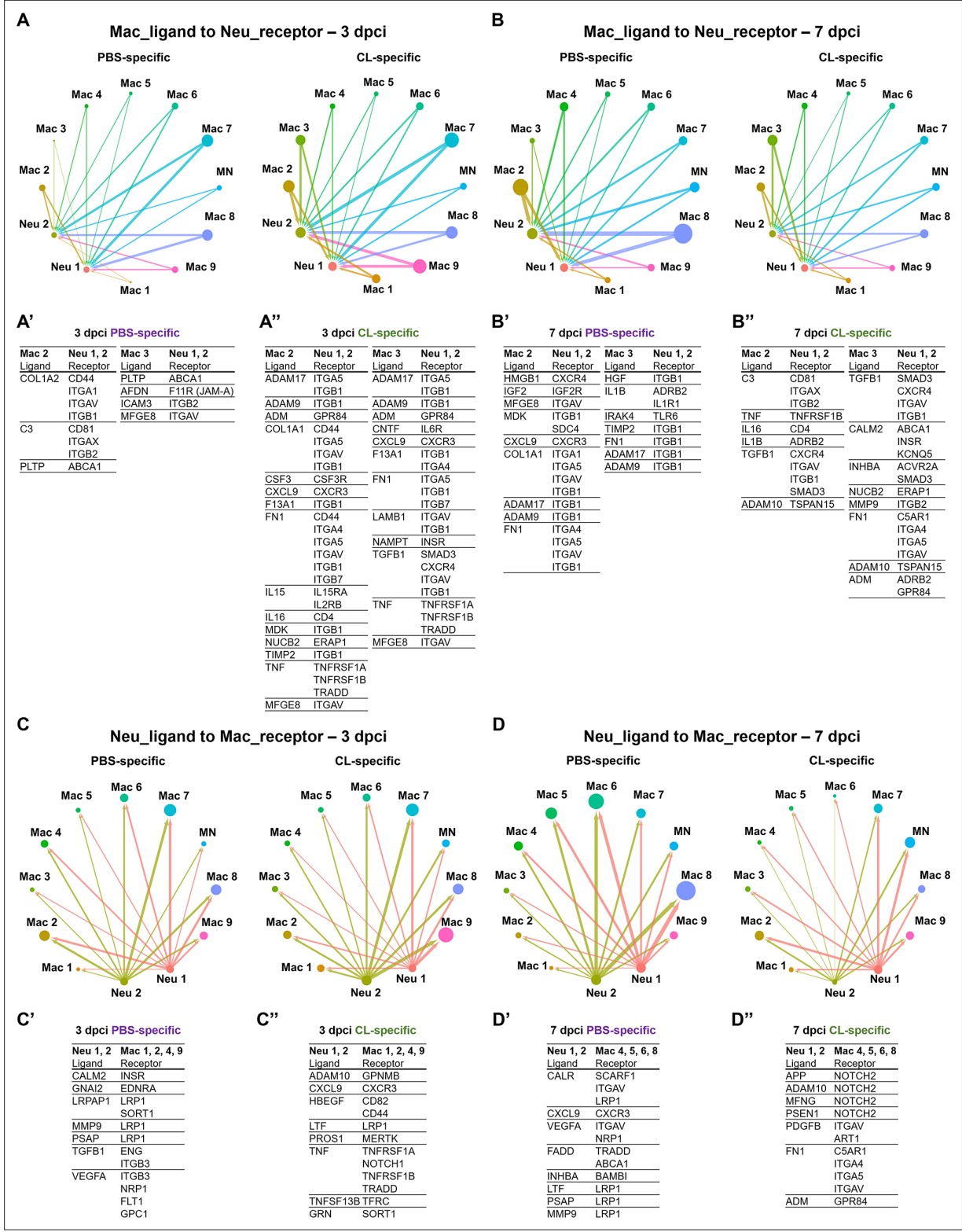

**Figure 5.** Cell–cell interactions between macrophages and neutrophils are altered in macrophage-delayed conditions. Crosstalk analyses identify hypothetical cell–cell interactions in macrophage and neutrophil clusters under regenerative (PBS) or macrophage-delayed (CL) conditions. (**A, B**) Putative interaction maps of macrophage-expressing ligands and neutrophil-expressing receptors among cell clusters at 3 days post cryoinjury (dpci) (**A**) and 7 dpci (**B**). Purple and green highlight the ligand–receptor pairs found specifically under PBS- or CL-treated conditions. Direction of arrows indicates the ligands signaling to the receptors in responding clusters. Circle size represents the numbers of ligand/receptor genes. Ligand–receptor

*Figure 5 continued on next page*

*Figure 5 continued*

pairs of the resident population Mac 2 and 3 to neutrophil clusters at 3 dpci (**A', A"**) and 7 dpci (**B', B"**) are shown. (**C, D**) Putative interaction maps of neutrophil-expressing ligands and macrophage-expressing receptors between clusters at 3 dpci (**C**) and 7 dpci (**D**). Purple and green highlight the ligand–receptor pairs found specifically under PBS- or CL-treated conditions. Direction of arrows indicates the ligands signaling to the receptors in responding clusters. Circle size represents the numbers of ligand/receptor genes. (**C', C"**) Ligand–receptor pairs of Neu 1, 2 to major macrophage responders at 3 dpci. (**D', D"**) Ligand–receptor pairs of Neu 1, 2 to major macrophage responders at 7 dpci.

The online version of this article includes the following source data for figure 5:

**Source data 1.** Macrophage ligand-to-neutrophil receptor pairs from celluar crosstalk analyses at 3 dpci.

**Source data 2.** Macrophage ligand-to-neutrophil receptor pairs from celluar crosstalk analyses at 7 dpci.

**Source data 3.** Neutrophil ligand-to-macrophage receptor pairs from celluar crosstalk analyses at 3 dpci.

**Source data 4.** Neutrophil ligand-to-macrophage receptor pairs from celluar crosstalk analyses at 7 dpci.

## Pseudotemporal trajectory analyses identify distinct progression routes and enriched genes among macrophage and neutrophil subpopulations

To reveal the potential progression from resident macrophages from naïve state to heterogeneous subpopulations post cardiac injury, we performed cell trajectory analysis using Monocle3 (*Figure 6*). We set the root from Mac 3, the main resident population identified in the uninjured hearts, which highly expresses progenitor/lineage markers *spi1b*, *coro1a*, and *irf8*. Macrophage clusters and neutrophil clusters were both subjected to the analysis due to the MN clusters constitute of both cell characteristics. As a result, macrophages occupied the pseudotime point 0 to pseudotime point 11 and arranged on the radial trajectories from Mac 3 while Cluster MN was the intermediary cluster along the trajectory to neutrophil population (*Figure 6A*). On the other hand, Neu 2 and Neu 1 occupied the pseudotime point 13 to pseudotime point 27 (*Figure 6A*). Four radial trajectories/routes were depicted, suggesting that macrophages were highly plastic and simultaneously polarized into heterogeneous subtypes with specialized functions during cardiac repair. The results also suggested that resident macrophages may play a sentinel role by detecting tissue damage and recruiting neutrophils to the injured site as previously reported (*Schiwon et al., 2014*). In combination with the profiling results, the resident Mac 3 mainly functions in immune system process and cell differentiation, suggesting that they might be involved in monocyte recruitment and differentiation into macrophages (*Table 1*). Route I represents transition from Mac 3 to Mac 1, which is mainly involved in iron homeostasis, angiogenesis, and anti-inflammatory activities with vigorous endocytosis activities under the regenerative condition (*Table 1* and *Figure 6A*). Route II represents transition from Mac 3 to pro-inflammatory clusters MN and shows the regenerative-associated polarization (*Figure 6A*) that MN clusters were constituted of more macrophage-like cells in regenerative hearts (purple route II) and more neutrophil-like cells in non-regenerative hearts (green route II). Route III represents transition from Mac 3 to proliferating Mac 5 via Mac 4, 6, and 7 for responding to oxidative stress, oxidative phosphorylation, and neutrophil infiltration (*Figure 6A*). Interestingly, the last route IV reveals transition from Mac 3 to Mac 2, and some of them shifted back with their gene signature similar to Mac 3, suggesting potential transition of these clusters during heart regeneration (*Figure 6A*). Coincidentally, Mac 2 is the most expanded cluster in regenerative condition (*Figure 3B*), while this plastic transition was absent in CL7d (*Figure 6B*). Next, we identified the DEG changes dynamically over the pseudotime of cardiac repair and analyzed the GO BP and KEGG pathways in which they are involved (*Figure 6C*). At early pseudotime points 0–4, genes related to phagocytosis (*marco*, *mrc1b*, and *havcr1*), complement system (*c1qc* and *c1qb*), and antigen sensing/presentation pathogen receptor (*cd74a*, *mhc2dab mfap4*) were highly enriched. These expression pattern fit nicely with the role of resident macrophages in cardiac homeostasis in removing damaged cells/components and serving as sentinel cells for antigen presentation to other immune cells (*Figure 6C*; *Nicolás-Ávila et al., 2020*; *Yan et al., 2013*). Later at pseudotime points 5–9, macrophages enhanced their phagocytosis and lysosomal capacity for debris clearance in response to DAMPs in order to prevent further tissue damages (*Figure 6C*). Interestingly, a small set of genes peak in the middle pseudotime points 10–14, which correspond to diverse function in regeneration (*grn1* and *grna*), oxygen transport (*habb1* and *hbba1.1*), cell proliferation (*top2a*, *mki67*, and *cdk1*), and inflammation/ribosome function (*tnfb*, *tlr5b*, and *rpls*) (*Figure 6C*). These results represent a diverse polarization of different macrophage subpopulations observed in respective routes (*Figure 6A*). Lastly, neutrophil-specific genes (*lect2l*, *tcnbb*, and *scpp8*) were highly expressed after

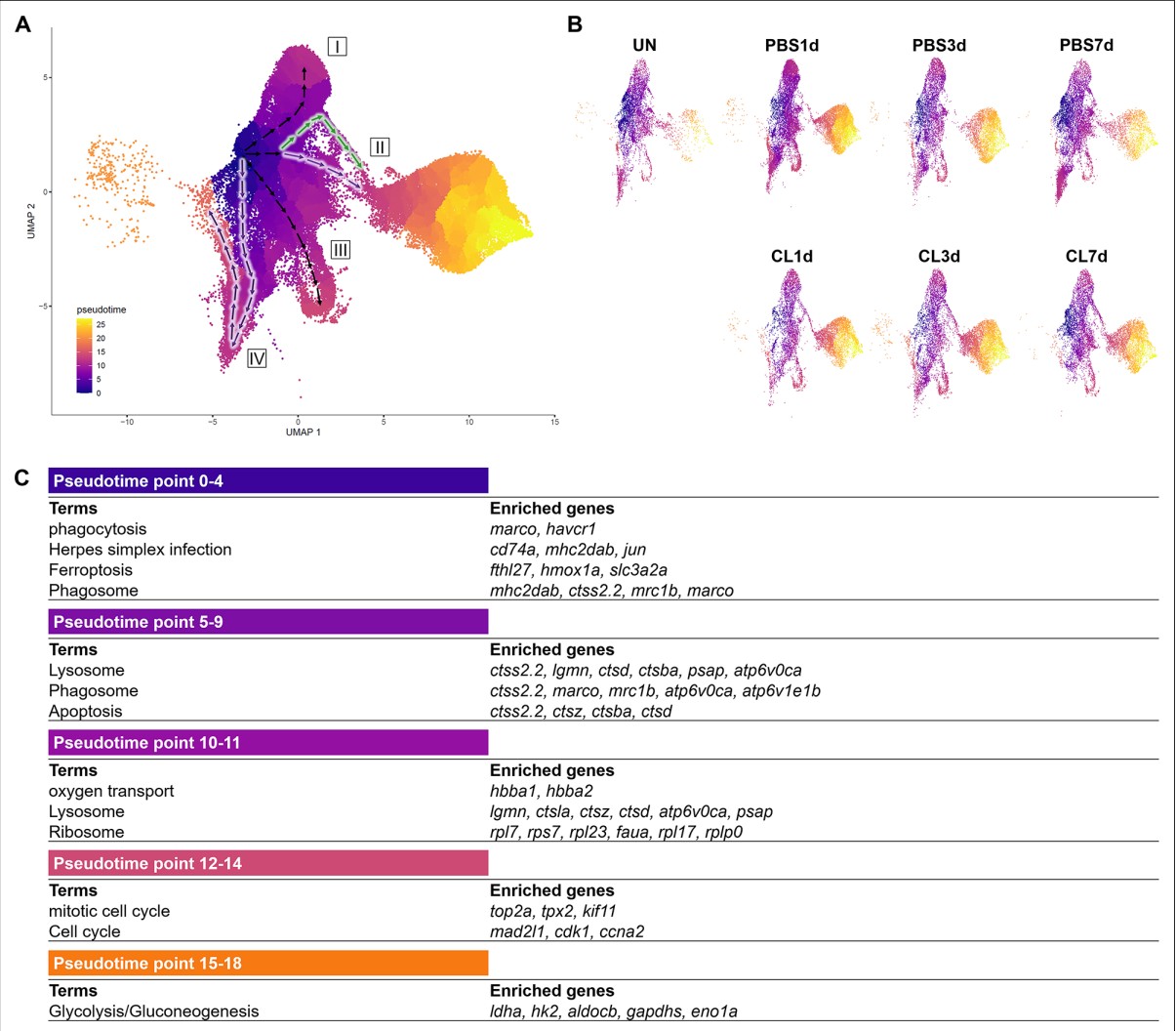

**Figure 6.** Pseudotemporal trajectory analyses identify distinct progression routes and enriched genes among macrophage and neutrophil subpopulations. (**A**) Pseudotemporal trajectory analyses of macrophages and neutrophils in zebrafish heart. Macrophages and neutrophils were subset using Seurat and input in Monocle3. The root of the pseudotime trajectory of macrophages was set based on resident macrophage cluster Mac 3 highly express progenitor genes *spi1* and *coro1*. Cells are colored by dark blue to bright yellow according to the earliest state to the latest state in pseudotime. Arrows indicate the direction of the cell-state transition through the pseudotime. Purple highlights the regenerative-associated direction, whereas green highlights the macrophage-delayed-associated direction. Directions of cell-state transition are indicated by Roman numerals. (**B**) Pseudotime of macrophages and neutrophils under each time point and condition is shown in the split UMAP plots. (**C**) GO and KEGG analysis of macrophage DEGs along pseudotime. DEGs, differentially expressed genes; GO, Gene Ontology; BP, biological process; KEGG, Kyoto Encyclopedia of Genes and Genomes; UMAP, Uniform Manifold Approximation and Projection.

The online version of this article includes the following source data for figure 6:

**Source data 1.** R script of pseudotemporal trajectory analysis.

pseudotime point 14. Glycolysis-related genes associated with inflammatory activities in neutrophils were consistently identified in Neu 2, while *npsn* and *lyz* were enriched at the latest pseudotime during neutrophil progression in Neu 1 (*Figure 6C*). Taken together, these results depict the main trajectories/routes of how macrophage might polarize and progress to gain different function during homeostasis and post cardiac injury, and highlight the regenerative route constitute of mainly resident clusters Mac 2 and 3.

## Depletion of resident macrophage compromises heart regeneration

Based on the single-cell profiling results, regeneration-associated macrophages were mainly resident macrophages Mac 2 and Mac 3, which were substantially enriched in regenerative PBS-control

hearts compared with non-regenerative –1d_CL-treated hearts. To test the functional significance of these resident macrophages without disrupting the circulation/monocyte-derived macrophage recruitment, we perform CL depletion earlier at 8 d prior to cardiac injury (–8d_CL, resident macrophage deficient model) (*Figure 7—figure supplement 1*). In our previous study, the effect of –1d_CL depletion on macrophage numbers only lasted a couple of days while their numbers soon recovered within a week afterward post cryoinjury (*Lai et al., 2017*). We first examined the macrophage content in *Tg(mpeg1:mCherry)* reporter fish by flow cytometry at 2 and 8 d post CL injection (2 dpip and 8 dpip; *Figure 7—figure supplement 1A*). Indeed, the proportion of mCherry⁺ cells in the zebrafish hearts dropped from 1.7% at steady state (green line) to 1.18% at 2 dpip (blue line), corresponding to the previous findings in –1d_CL-treated hearts (*Figure 7—figure supplement 1A*; *Lai et al., 2017*). At 8 dpip, the proportion of mCherry⁺ cells recovered to 1.64%, comparable to the steady-state level (*Figure 7—figure supplement 1A*, orange line). However, these recovered macrophages express *mpeg1*:mCherry at a lower level, suggesting their differentiation status freshly from circulating progenitors/monocytes (*Figure 7—figure supplement 1A*, orange line). These results also suggest that not all the resident *mpeg1*:mCherry⁺ cells are susceptible to CL depletion, especially for those B cells we observed from single-cell profiling (*Figure 3A*). Next, we performed cryoinjury on these fish after –8d_CL treatment and examined the macrophage content at 1 dpci (*Figure 7—figure supplement 1B*). Surprisingly, the proportion of mCherry⁺ cells was even higher in the –8d_CL group (1.14%) compared to PBS-controls (0.83%). Since scRNAseq results showed specific loss of Mac 2 and Mac 3, we examined their cluster-specific gene *hbaa1* and *timp4.3* expression by qPCR to test whether the Mac 2 and Mac 3 could recover from –8d_CL depletion. Consistent with our scRNAseq results, the expression of both genes in *mpeg1*:mCherry⁺ cells was significantly reduced in –8d_CL hearts compared to PBS-controls (*Figure 7—figure supplement 1C*). ISH also showed drastic reduction of *timp4.3*⁺ cells in –8d_CL hearts than in PBS-controls (*Figure 7—figure supplement 1D*). These results support that early CL administration (–8d_CL) depletes resident macrophages without affecting overall macrophage infiltration after cardiac injury. Replenished macrophages after depletion may potentially derive from circulating monocyte, but resident macrophage Mac 2 and Mac 3 remained diminished without replenishment from those *mpeg1*:mCherry-low monocyte-derived cells (*Figure 7—figure supplement 1A*). Of note, we observed an overshoot of macrophage infiltration to the –8d_CL injured hearts, consistent with previously published results showing that –1d_CL treatment actually led to more macrophages infiltrated in the injured heart at 7 dpci (*Lai et al., 2017*). This observation suggests an intrinsic role of resident macrophages in modulating inflammation and immune cell recruitment after cardiac injury.

To determine the functional requirement of resident macrophages in cardiac regeneration, we then examined the key processes of successful heart regeneration, including revascularization, CM proliferation, and scar resolution in –8d_CL and PBS-control hearts (*Figure 7A*). Consistent with our previous study, we used *Tg(fli1:EGFP;myl7:DsRed-NLS)* fish to visualize the vascular endothelial cells in green and the nuclei of CMs in red, and used Acid Fuchsin-Orange G (AFOG) staining to reveal the scar size and composition in the injured hearts (*Lai et al., 2017*). Fast revascularization of the injured area in the first week post injury is essential to support zebrafish heart regeneration and was compromised in –1d_CL hearts (*Lai et al., 2017*; *Marín-Juez et al., 2016*). We examined revascularization by ex vivo imaging of GFP⁺ endothelial cells in whole-mount hearts and found that revascularization of the injured area in –8d_CL hearts was significantly decreased than PBS-controls at 7 dpci (*Figure 7A* and *Figure 7—source data 1*). Neutrophil numbers were also higher in the injured area of –8d_CL hearts, suggesting a similar neutrophil retention phenotype (*Figure 7B* and *Figure 7—source data 1*). We next examined and quantified proliferating CMs by EdU incorporation since dedifferentiation and proliferation of the existing CMs are the major source to replenish the lost myocardial tissue from injury (*Sallin et al., 2015*). Unlike previously reported in –1d_CL hearts, we did not observe a significant decrease of CM proliferation in –8d_CL hearts (*Figure 7C* and *Figure 7—source data 1*). Instead, we noticed small and round CM nuclei with weaker DsRed signals in the border zone of –8d_CL hearts compared to controls (*Figure 7C*; I–IV). Correspondingly, the density of CM nuclei was significantly lower in the border zone of the –8d_CL hearts (*Figure 7C*). Furthermore, injured areas were also larger in –8d_CL hearts (*Figure 7C*). Together, these results suggest that less CMs survived from the initial injury and were able to re-enter cell cycle and proliferate. To test this possibility, we performed TUNEL assay to label damaged nuclei of dying cells. Indeed, we found significantly more

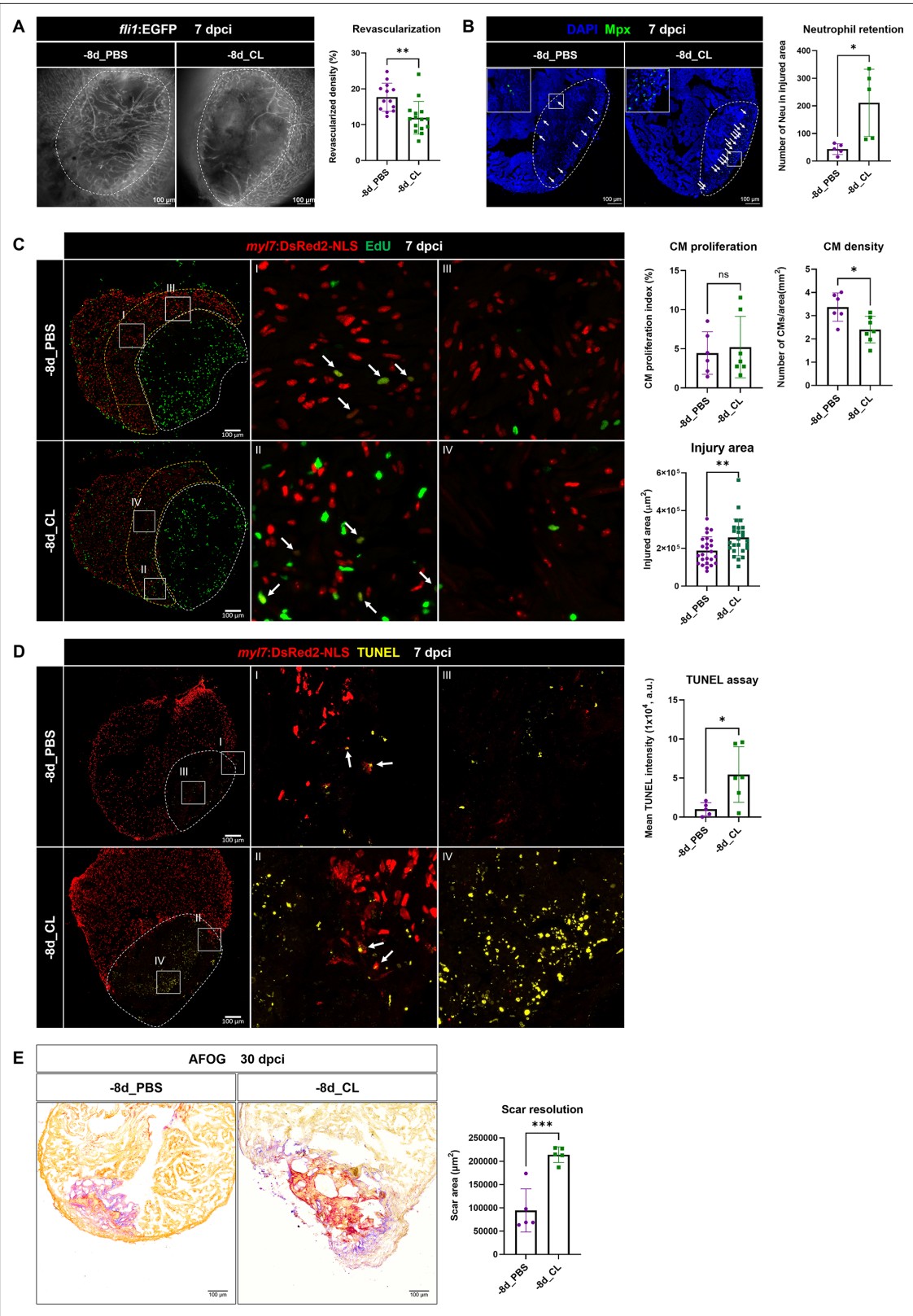

**Figure 7.** Depletion of resident macrophages compromised heart regeneration. Functional validation of resident macrophage depletion in cardiac repair. Zebrafish were IP-injected with PBS or CL at 8 d before cardiac injury (–8d_PBS or –8d_CL). (**A**) Revascularization was evaluated at 7 days post cryoinjury (dpci). Endogenous *fli1*:EGFP fluorescence depicts the vascular endothelial cells. White dotted lines delineate injury areas; scale bars, 100 µm. Quantification of vessel density by ImageJ is shown in the right panel (n ≥ 13; p=0.0015). (**B**) Neutrophils in injured areas were examined

*Figure 7 continued on next page*

*Figure 7 continued*

by Myeloperoxidase (Mpx) immunostaining. White dotted lines delineate injury areas; scale bars, 100 μm. Quantification of neutrophil number in injured areas is listed in the right panel (n = 5; p=0.0359). (**C**) CM proliferation was assessed by EdU cell proliferation assay at 7 dpci. White dotted lines delineate injury areas; scale bars, 100 μm (left panels). White arrows point out the proliferating CMs (insets I and II). *myl7*:DsRed-NLS served as the endogenous CM nuclear marker. The shape of CMs became smaller and DsRED fluorescence was weaker in CL-pretreated hearts (–8d_PBS vs. –8d_CL, insets III vs. IV). Quantification of the injured area (n = 25; p=0.0061), along with the CM density (n ≥ 6; p=0.0131) and CM proliferation index (n ≥ 6; p=0.6990) in 200 μm adjacent to the injured area (border zone, delineated by yellow dotted lines) are shown in right panels. (**D**) TUNEL assay was performed on the same batch of cryosections, which identified the CMs lost in the border zone (insets I and II) at 7 dpci. White arrows point out the TUNEL-positive CMs in the area; scale bars, 100 μm (left panels). More damaged nuclei were found in CL-treated hearts than in PBS controls (inset III and IV). Quantification of TUNEL intensity is listed in the right panel (n ≥ 5; p=0.0269). (**E**) Scar resolution was evaluated by Acid Fuchsin Orange G (AFOG) staining at 30 dpci. AFOG staining visualized healthy myocardium in orange, fibrin in red, and collagen in blue. Quantification of scar area is shown in the right panel (n = 5; p=0.0006). CL, clodronate liposomes; IP, intraperitoneal; CM, cardiomyocyte. The heart samples under regenerative (–8d_PBS) or resident macrophage-deficient (–8d_CL) conditions are indicated by purple or green, respectively. Student's *t*-test was used to assess all comparisons by Prism 9.

The online version of this article includes the following source data and figure supplement(s) for figure 7:

**Source data 1.** Statistic analyses of neovascularization, neutrophil numbers, CM proliferation and density, scar area of the PBS vs. -8d_CL-treated hearts.

**Source data 2.** Statistic analyses of *timp4.3* and hbaa expression by qPCR.

**Source data 3.** Statistic analyses of neovascularization, neutrophil numbers, CM proliferation and density, scar area of the PBS vs. -1m_CL-treated hearts.

**Source data 4.** *timp4.3*[+] resident macrophage depletion in CL vs. PBS control hearts at 8 days after IP injection.

**Figure supplement 1.** Specific resident macrophage clusters Mac 2 and 3 were non-recoverable after CL-mediated depletion.

**Figure supplement 2.** Depletion of resident macrophages led to long-term incompetence of heart regeneration.

TUNEL-positive cells within the injured area of –8d_CL hearts (*Figure 7D* and *Figure 7—source data 1*). As a result, –8d_CL hearts exhibited larger/unresolved scar tissues composed of both collagen and fibrin than –8d_PBS hearts at 30 dpci, reflecting compromised heart regeneration (*Figure 7E* and *Figure 7—source data 1*).

Since we observed a replenishment of macrophages after –8d_CL treatment (*Figure 7—figure supplement 1A*), we further tested whether the regenerative capacity may be restored after longer recovery and perform cryoinjury 1 mo after CL depletion (–1 m_CL, *Figure 7—figure supplement 2* and *Figure 7—source data 2*). Strikingly, –1 m_CL hearts still failed in regeneration, exhibiting significant defects in revascularization, neutrophil retention, and scar resolution (*Figure 7—figure supplement 2*). These results suggest that resident macrophages are prerequisite for heart regeneration in modulating the revascularization, CM survival, and the resolution of inflammation and fibrotic scars, which cannot be replenished or functionally replaced by circulation/monocyte-derived macrophages upon depletion.

## The resident macrophage population Mac 2 expresses *hmox1a*, involved in heme clearance

To gain mechanistic insight into how resident macrophages participate in heart regeneration, particularly in CM survival, we further characterized *hmox1a*, a DEG in resident macrophage Mac 2 (*Figure 8A*). Heme released by damaged erythrocytes (hemoglobin) and CMs (myoglobin) is a major source of ROS stress after tissue injury, often leading to secondary damage and extended cell death (*Chiang et al., 2018*). *hmox1a* encodes Heme oxygenase-1, which degrades heme into cardioprotective and anti-inflammatory effectors such as biliverdin/bilirubin, carbon monoxide (CO), and free iron/ferritin (*Tomczyk et al., 2019*). Biliverdin/bilirubin and CO further exert antioxidant and anti-inflammatory activities, upregulate genes such as *il10*, and promote M2 macrophage polarization (*Vijayan et al., 2018*). Heme oxygenase-1 may also regulate iron homeostasis and ferroptosis, the main type of cell death after MI (*Chiang et al., 2018*; *Vijayan et al., 2018*). We first validated the expression of *hmox1a* in both –8d_PBS and –8d_CL hearts at 7 dpci by ISH and observed the *hmox1a*[+] cells with macrophage-like morphology. These *hmox1a*[+] cells were located in the injured area and around the epicardial layer in the –8d_PBS-control hearts, and largely reduced with weaker expression in resident macrophage-deficient hearts (*Figure 8B*). We further confirmed their expression in macrophage by immunostaining against Hmox1 in the *mpeg1:mCherry* background (*Figure 8C*). Corresponding to

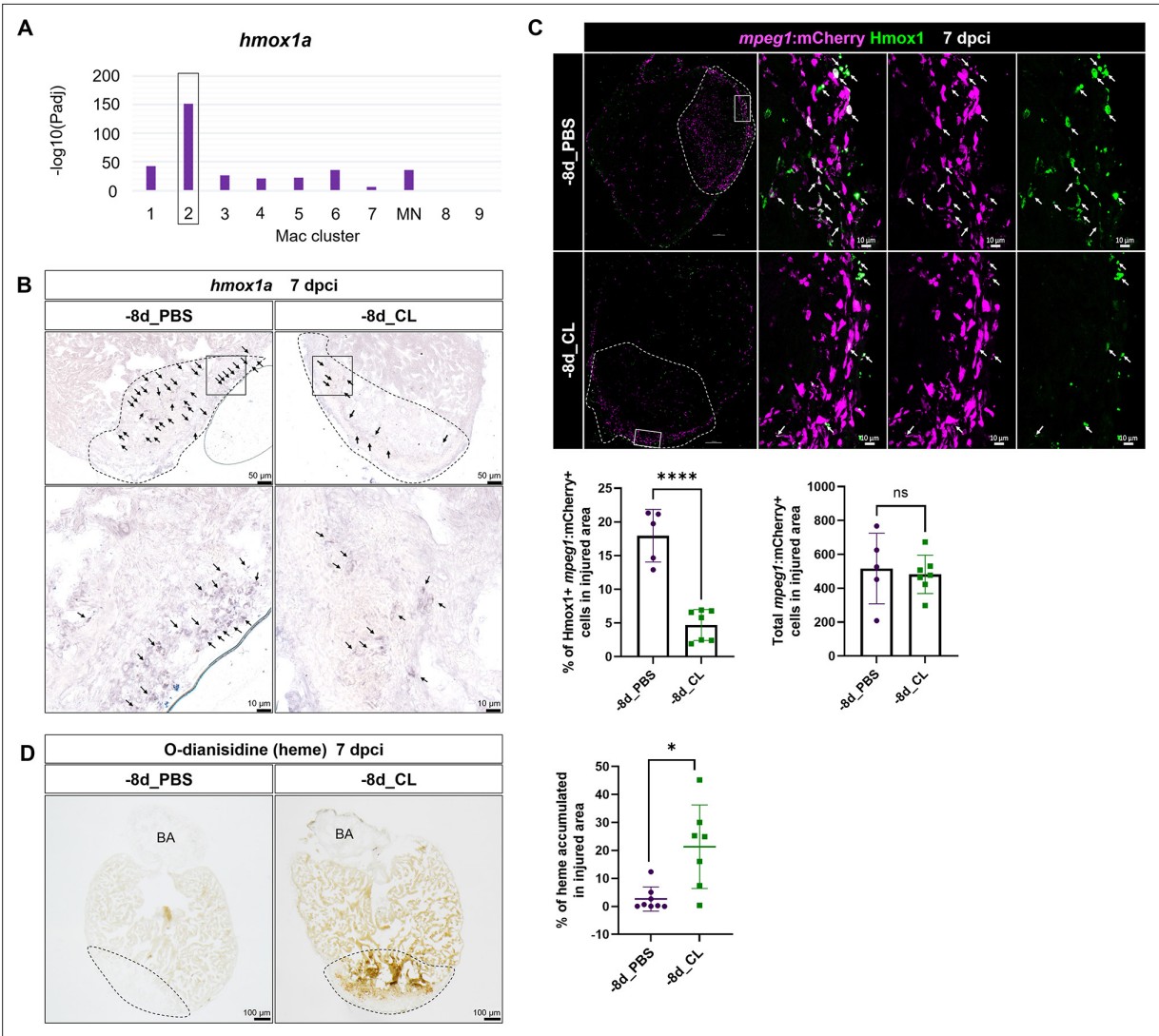

**Figure 8.** Resident macrophages Mac 2 express *hmox1a* for heme clearance during cardiac repair. Reduced *hmox1a*-expressing macrophages were associated with heme accumulation under resident macrophage-deficient condition. (**A**) The bar plot indicates the *hmox1a* enrichments (–log$_{10}$ adjusted p-value) across macrophage clusters. (**B**) Expression of *hmox1a* was detected by in situ hybridization in the regenerative (–8d_PBS) and macrophage-deficient (–8d_CL) hearts at 7 days post cryoinjury (dpci). The dotted lines delineate injury areas; scale bars, 50 µm (upper panels) and 10 µm (lower panels). (**C**) Hmox1-expressing macrophages (arrows) were examined by immunostaining in the regenerative hearts (–8d_PBS) or resident macrophage-deficient hearts (–8d_CL) at 7 dpci. Quantification of the percentage of Hmox1$^+$/*mpeg1*:mCherry$^+$ macrophages and the number of total *mpeg1*:mCherry$^+$ macrophages in injured area are shown in lower panels (left, Hmox1$^+$/*mpeg1*:mCherry$^+$ macrophages, n ≥ 5; p<0.0001; right, total *mpeg1*:mCherry$^+$ macrophages, n ≥ 5; p=0.72). White dotted lines delineate injury areas; scale bars, 100 µm (left panels) and 10 µm (right panels). (**D**) O-dianisidine staining of regenerative hearts (–8d_PBS) and resident macrophage-deficient hearts (–8d_CL) at 7 dpci. The dotted lines delineate injury areas; scale bars, 100 µm. Quantification of staining density by ImageJ is shown in the right panel (n ≥ 7; p=0.0155). The heart samples under regenerative (–8d_PBS) or resident macrophage-deficient (–8d_CL) conditions are indicated by purple or green, respectively. Student's *t*-test was used to assess all the comparisons by Prism 9.

The online version of this article includes the following source data for figure 8:

**Source data 1.** Statistic analyses of Hmox1a+/mpeg1:mCherry+ macrophages and o-dianosidine positive area in -8d-CL vs. PBS control hearts.

the scRNAseq profiling results and the ISH results, a portion of *mpeg1*:mCherry$^+$ macrophages indeed expresses Hmox1 in the –8d_PBS-control hearts, while double-positive cells were significantly less in the resident macrophage-deficient hearts (***Figure 8C***). Since Hmox1 functions in metabolizing heme, we examined heme accumulation in the injured hearts by O-dianisidine staining (***Figure 8D***). Indeed, heme released in the injured area was largely cleared in the –8d_PBS-control hearts at 7 dpci as a part of debris clearance and microenvironment remodeling during cardiac repair. However, excessive heme

accumulated in the injured area of the resident macrophage-deficient hearts, potentially affecting the CMs' survival as we observed previously (*Figures 7C and 8D*). These results suggest that resident macrophages accelerate cardiac repair partly by clearing heme and preserve the regenerative CMs from ROS stress and extended cell death.

## Discussion

### Timely inflammatory resolution and metabolic switch are critical events associated with macrophages function during zebrafish heart regeneration

Previously, Lai et al. showed that delaying macrophage recruitment in the first week post cardiac injury by CL-mediated predepletion compromised zebrafish heart regeneration in terms of revascularization, neutrophil retention, CM proliferation, and scar resolution, even though the macrophage numbers gradually recovered before 7 dpci (*Lai et al., 2017*). These data indicated a certain degree of functional divergence in the late infiltrating macrophages compared with the macrophages in the PBS-control hearts. Dynamics of inflammatory cells were also investigated during cardiac repair in mammals. Upon MI, reperfusion is the standard practice in clinics, which salvages some CMs from ischemic death and thus reduce the infarcted area. In the mouse ischemic-reperfusion (IR) model, the vessel occlusion is released 30 min after ligation to allow blood flow and immune cell trafficking compared to the permanent ligation (MI) model. Interestingly, inflammatory cell dynamic in the ischemic-reperfusion model resembles the regenerative zebrafish, when macrophage infiltration peak at an earlier time point, and neutrophils also resolve faster than in the permanent MI model, even though the cell numbers might not be comparable due to the differences in cell death and infarct size (*Yan et al., 2013*). Furthermore, macrophages from the IR model seem to repolarize and become inflammation-resolving M2 type faster than those from the permanent MI model, suggesting a progressive inflammation resolution (*Yan et al., 2013*). In the present study, we further characterized the PBS vs. –1d_CL models in zebrafish and found the transcriptomic changes upon macrophage delay. –1d_CL-treated hearts showed prolonged inflammation and dysregulated metabolism even until 21 dpci, suggesting that macrophages may play important roles in modulating neutrophil and inflammation resolution, which was further supported by the scRNAseq profiling of these inflammatory cells (discussed below). However, whether and how macrophages may regulate cardiac metabolism remain unclear. Metabolic shift in utilizing different metabolites for energy production is a critical process during cardiac repair and regeneration (*Ellen Kreipke et al., 2016*; *Doenst et al., 2013*; *Zuurbier et al., 2020*). Compared with PBS-control hearts, we observed downregulated genes predicted to be involved in both glycolysis and oxidative phosphorylation in CL-treated hearts from 7 to 21 dpci. The aberrant metabolism might relate to macrophage phagocytosis function since cardiac macrophages have been shown to preserve the metabolic stability of CMs by actively clearing the dysfunctional mitochondria and other waste via phagocytosis during homeostasis (*Nicolás-Ávila et al., 2020*), while mitochondrial dysfunction leads to impaired revascularization and fibrosis formation in mammals (*Satoh et al., 2011*; *Yu et al., 2019*; *Li et al., 2020*). Whether similar events occur during cardiac repair and how macrophages may influence CM metabolism during dedifferentiation, proliferation, and redifferentiation await future investigation.

### ScRNAseq profiling revealed heterogeneous cardiac resident macrophages during steady-state and repair/regeneration

Since macrophages and neutrophils seem to be the main players in the regenerative PBS vs. non-regenerative –1d_CL conditions, we further profiled these inflammatory cells by scRNAseq. During cell isolation, we noticed a portion of *mpeg1*:mCherry⁺/*mpx*:EGFP⁺ cells in steady state and enriched in PBS-treated hearts post injury. These double-positive cells were confirmed as macrophages by cell sorting and Giemsa staining and were also described previously in the fin-fold amputated larvae (*Mathias et al., 2009*; *Mathias et al., 2006*). From the scRNAseq profiling, we only observed a corresponding population of *mpeg1*⁺/*mpx*⁺ macrophages in Mac 2 at PBS7d. We speculate that EGFP proteins are stable in sorted *mpeg1*:mCherry⁺/*mpx*:EGFP⁺ macrophages and the *mpx* gene only needs to be turned on after injury and the replenishment of Mac 2 in PBS-treated heart. Myeloperoxidase-expressing macrophages were observed in human atherosclerotic plaque and increased during

inflammation (*Sugiyama et al., 2001*). Among macrophage clusters, Mac 1, 2, 3, 4, and 8 are the major subsets found in the naïve/untouched hearts (*Figure 3B*). The cluster-enriched genes show similar functions to murine cardiac resident macrophages, including phagocytosis/homeostasis, angiogenesis, antigen presentation, and sentinel function in monocytes and neutrophil chemotaxis (*Dick et al., 2019*; *Alvarez-Argote and O'Meara, 2021*). However, the cluster/lineage markers were not comparable between fish and murine cardiac resident macrophages. The origin/lineage of fish cardiac macrophage is also difficult to determine without comparable markers and lineage-tracing tools as in rodents. Therefore, new genetic tools are needed for further investigating these cardiac macrophage clusters. Nevertheless, judging by the dynamic change post injury and their replenishment in CL-treated hearts, Mac 2, 3, and 8 might have originated from the embryonic lineage as they show minor or no replenishment after depletion and injury. On the other hand, Mac 1 and 4 showed active recruitment to the injured heart and stronger *ccr2* expression (Mac 1), suggesting that they might be derived from monocytes (*Figure 3B*). After cardiac injury, the dynamic changes in numbers and gene expression among heterogeneous macrophage clusters reflect the swift response to injury and high plasticity of polarization, making it difficult to define regeneration-associated macrophage properties. Thus, we combined multiple analyses, including cell proportional analysis, GO analysis on cluster-enriched and condition-enriched DEGs, and crosstalk analysis in the present study. Interestingly, we observed some macrophage clusters other than Mac 2 and 3 turn on and off specific sets of genes under regenerative vs. non-regenerative conditions without affecting their overall identity (in clustering, *Figure 4*). This phenomenon of gene alternative activation suggests that functional polarization of circulation/monocyte-derived macrophages might be regulated by resident macrophages. Interestingly, in addition to retention, neutrophils also exhibited alternative property upon macrophage pre-depletion, with tissue repairing/regeneration associated and self-clearance genes expression under regenerative condition and inflammatory exacerbation genes under regenerative condition. These results highlight the importance of interactions between neutrophils and both resident and recruited macrophages and how they coordinately modulate inflammation and tissue repair.

## Zebrafish cardiac resident macrophages are indispensable for heart regeneration

Our functional results reveal the essential roles of cardiac resident macrophages in zebrafish heart regeneration as they are required to support revascularization, CM survival and replenishment, and eventually scar resolution. However, we did not observe cardiac resident macrophages directly contributing to CM dedifferentiation and proliferation. CM repopulation is affected possibly due to the reduced pool of regenerative cells (*Figure 7C*), compromised revascularization (*Marín-Juez et al., 2019*), and the abnormal ECM remodeling, which leads to scar deposition without affecting CM proliferation (*Wang et al., 2013*; *Allanki et al., 2021*). From our results, we identified that resident Mac 2 expresses genes in regulating ROS homeostasis, including *hmox1a*, which functions in heme degradation and alleviating ROS stress for CM survival. In addition, we found that resident macrophages express ECM components and remodeling genes, including fibronectin, collagen, ADAM metallopeptidases, and tissue inhibitor of metalloproteinase. Dynamic remodeling of ECM is associated with immune cell behavior, fibrosis, tissue stiffness for CM proliferation and migration, and inflammation, which together provide a pro-regenerative microenvironment for cardiac repair. On the other hand, alternative activation of monocyte-derived Mac 1 may contribute to CM proliferation directly as we found genes associated with angiogenesis and cardiovascular development under regenerative condition. Particularly, *vegf* has been associated with CM proliferation in both larval and adult heart regeneration (*Figure 4*; *Bruton et al., 2022*; *Pronobis and Poss, 2020*). Methodologically, CL was administrated by IP, so we also cannot rule out the potential systemic effects of losing CL-sensitive cells elsewhere in the body on cardiac repair, despite having examined the macrophage recruitment to the injured hearts after a long recovery period. In addition, resident macrophages are usually confirmed through lineage-tracing experiments to understand their embryonic origin. Future studies using more sophisticated methods are required to confirm the embryonic origin of resident macrophages and functionally validate the specific mechanisms of how these resident macrophages may directly or indirectly contribute to CM proliferation and zebrafish heart regeneration.

In summary, our study characterized the plasticity and heterogeneity of both macrophages and neutrophils at steady state and during the first week post injury. In addition, we identified cardiac

resident macrophages and functionally characterized their roles in heart regeneration. New genetic tools and models are required to further investigate the origin and function of distinct macrophage subsets and the molecular mechanism underlying how cardiac macrophages contribute to heart regeneration, which may gain knowledge and provide new insights for developing therapeutic strategies for cardiac repair.

# Materials and methods

**Key resources table**

| Reagent type (species) or resource | Designation | Source or reference | Identifiers | Additional information |
|---|---|---|---|---|
| Strain, strain background (*Danio rerio*, AB) | *Tg(mpeg1.4:mCherry-F)ump2* | PMID:24567393 | N/A | |
| Strain, strain background (*D. rerio*, AB) | *TgBAC(mpx:GFP)i114* | PMID:16926288 | N/A | |
| Strain, strain background (*D. rerio*, AB) | *Tg(–5.1myl7:DsRed2-NLS)f2Tg* | PMID:12464178 | N/A | |
| Strain, strain background (*D. rerio*, AB) | *Tg(fli1a:EGFP)y1* | PMID:12167406 | N/A | |
| Antibody | Anti-Mpx (rabbit polyclonal) | GeneTex | Cat# GTX128379; RRID:AB_2885768 | IF (1:500) |
| Antibody | Anti-mCherry (chicken polyclonal) | Abcam | Cat# ab205402; RRID:AB_2722769 | IF (1:250) |
| Antibody | Anti-HMOX1 (rabbit polyclonal) | Aviva | Cat# ARP45222_P050; RRID:AB_2046270 | IF (1:100) |
| Antibody | Anti-Digoxigenin Fab fragments antibody, AP conjugated (sheep polyclonal) | Roche | Cat# 11093274910; RRID:AB_514497 | ISH (1:1000) |
| Sequence-based reagent | *hbaa1*_F | This paper | qPCR primer | AAGCCCTCGCCAGAATGC |
| Sequence-based reagent | *hbaa1*_R | This paper | qPCR primer | ACCCATGATAGTCTTTCCGTGC |
| Sequence-based reagent | *timp4.3*_F | This paper | qPCR primer | GCAGAGACGCGGAGGTGAAG |
| Sequence-based reagent | *timp4.3*_R | This paper | qPCR primer | CGGGACCACAGCTACAAGCC |
| Sequence-based reagent | *eef1a1l1*_F | PMID:32359472 | qPCR primer | CTGGAGGCCAGCTCAAAC |
| Sequence-based reagent | *eef1a1l1*_R | PMID:32359472 | qPCR primer | ATCAAGAAGAGTAGTACCGCTAGCATTAC |
| Sequence-based reagent | *hbaa1*_F | This paper | ISH primer | ACGCAGCGATGAGTCTCT |
| Sequence-based reagent | *hbaa1*_R | This paper | ISH primer | CATGCATGAGTTGTTAAGAGTG |
| Sequence-based reagent | *timp4.3*_F | This paper | ISH primer | CAGACACGAAGGACATGC |
| Sequence-based reagent | *timp4.3*_R | This paper | ISH primer | CACCGAATGTATGTGTTTATTAAC |

*Continued on next page*

*Continued*

| Reagent type (species) or resource | Designation | Source or reference | Identifiers | Additional information |
|---|---|---|---|---|
| Sequence-based reagent | *hmox1a_F* | This paper | ISH primer | TCAGAGCATTCGAGTTCAAC |
| Sequence-based reagent | *hmox1a_R* | This paper | ISH primer | ACAGTTTATTAATCTTGCATTTACACAG |
| Commercial assay or kit | Click-iT EdU Cell Proliferation Kit for Imaging, Alexa Fluor 647 dye | Thermo Fisher | Cat# C10340 | |
| Commercial assay or kit | In Situ Cell Death Detection Kit, TMR red | Sigma-Aldrich | Cat# 12156792910 | |
| Commercial assay or kit | miRNeasy micro Kit | QIAGEN | Cat# 217084 | |
| Commercial assay or kit | Chromium single cell 3' Reagent kits v3 | 10x Genomics | N/A | |
| Commercial assay or kit | NBT/BCIP stock solution | Roche | Cat# 11681451001 | ISH (1:50) |
| Chemical compound, drug | Clodronate liposome | Liposoma | Cat# C-005 | 10 µl per fish |
| Chemical compound, drug | DIG RNA labeling mix | Roche | Cat# 11277073910 | ISH |
| Software, algorithm | NOIseq | PMID:26184878 | RRID:SCR_003002 | |
| Software, algorithm | WebGestalt | PMID:15980575 | RRID: SCR_006786 | |
| Software, algorithm | CellRanger | 10x Genomics | RRID: SCR_023221 | |
| Software, algorithm | Seurat | PMID:29608179 | RRID: SCR_007322 | |
| Software, algorithm | Monocle3 | PMID:30787437 | RRID: SCR_018685 | |
| Software, algorithm | Ligand-Receptor Interaction Network Analysis | PMID:26198319 | N/A | |
| Software, algorithm | Cytoscape | PMID:14597658 | RRID: SCR_003032 | |

## Experimental models

All zebrafish were maintained in our in-house fish facility at Institute of Biomedical Sciences, Academia Sinica, following standard husbandry protocol. All experiments were done following institutional and ethical welfare guidelines and animal protocols approved by the ethics committee of Academia Sinica. We crossed *Tg(mpeg1.4:mCherry-F)*[ump2] (*Bernut et al., 2014*) and *TgBAC(mpx:GFP)*[i114] (*Renshaw et al., 2006*) to generate *Tg(mpx:GFP;mpeg1.4:mCherry-F)* line as well as *Tg(−5.1myl7:DsRed2-NLS)*[f2Tg] (*Rottbauer et al., 2002*) and *Tg(fli1:EGFP)*[y1] *Lawson and Weinstein, 2002*) to generate *Tg(fli1:EGFP;−5.1myl7:DsRed2-NLS)* line respectively for our animal experiments. Intraperitoneal (IP) injections of 10 µl PBS and clodronate liposomes (5 mg/ml) (Liposoma, Amsterdam, The Netherlands) in each fish were performed according to the experimental design.

## Cryoinjury

Cryoinjury was performed as previously described in zebrafish (*Chablais et al., 2011*; *González-Rosa et al., 2011*; *Schnabel et al., 2011*). In brief, fish were anesthetized in 0.04% tricaine (Sigma, St Louis, MO) and immobilized in a damp sponge with ventral side up. A small incision was created through the

thoracic wall by microdissection scissors. A stainless steel cryoprobe precooled in liquid nitrogen was placed on the ventricular surface until thawing. Fish were then moved back to a tank of freshwater for recovery, and their reanimation could be enhanced by pipetting water onto the gills after surgery.

## Cryosections and histological analyses

Zebrafish hearts were extracted and fixed in 4% (wt/vol) paraformaldehyde (Alfa Aesar, MA) at room temperature for 1 hr. Collected hearts were subsequently cryopreserved with 30% (wt/vol) sucrose, followed by immersed in OCT (Tissue-Tek, Sakura Finetek, Torrance, CA) and stored at −80°C immediately. 10 µm cryosections were collected for histological analysis. AFOG staining was applied for the visualization of healthy CMs in orange, collagens in blue and fibrins in red. In brief, samples were incubated in preheated Bouin's solution (Sigma) at 58°C for 2 hr post fixation and subsequently immersed in 1% phosphomolybdic acid (Sigma) and 0.5% phosphotungstic acid solution (Sigma) at room temperature for 5 min for mordanting. Samples were then incubated with AFOG solution containing Aniline Blue (Sigma), Orange G (Sigma), and Acid Fuchsin (Sigma) for color development. Quantification was done by measuring the scar area in each heart from five discontinuous sections including the one with the largest scar as well as the two sections at the front and the two sections at the back as previously described (*Lai et al., 2017*). Statistic was performed on Prism 9 using Student's *t*-test.

## Quantitative polymerase chain reaction (qPCR)

For collecting single cells from hearts, 45–60 ventricles were isolated in each experiment at respective time points and conditions, pooled together and digested using LiberaseDH (Sigma) at 28°C in a 24-well plate. Cell suspension was filtered through 100 µm, 70 µm, and 40 µm cell strainers (SPL, Gyeonggi-do, Korea) and centrifuged at 200 × *g* for 5 min. Cell pellet was resuspended in 0.04% BSA/PBS, stained with DAPI, and filtered through 35 µm Flowmi cell strainer (BD, NJ). Cells were sorted using fluorescence-activated cell sorting (FACS; BD Facs Aria) and then subjected to RNA isolation. RNA was extracted using TRIzol LS Reagent (Life Technologies Invitrogen, CA) according to the manufacturer's instructions. First-strand cDNA was synthesized using SuperScript III First-Strand Synthesis System (Life Technologies Invitrogen) with oligo (dT)$_{18}$ primer. The qPCR analysis was carried out using DyNAmo ColorFlash SYBR Green qPCR Kit (Thermo Scientific, USA) on a LightCycler 480 Instrument II (Roche). The relative gene expression was normalized using *eef1a1l1* as an internal control. Oligonucleotide sequences for qPCR analysis are listed in *Figure 7—source data 2*.

## Immunostaining and imaging

For immunofluorescence, slides were washed twice with PBS and twice with ddH$_2$O, and then incubated in blocking solution (1×PBS, 2% [vol/vol] sheep serum, 0.2% Triton X-100, 1% DMSO). Slides were then incubated in primary antibodies overnight at 4°C, followed by three PBST (0.1% Triton X-100 in 1× PBS) washes and incubation with secondary antibodies for 1.5 hr at 28°C. Slides were washed again with PBST and stained with DAPI (Santa Cruz Biotechnology, TX) before mounting. Antibodies and reagents used in this study included anti-Mpx (GeneTex, San Antonio, TX) at 1:500, anti-mCherry (Abcam, Cambridge, UK) at 1:250 and anti-HMOX1 (Aviva, CA) at 1:100. EdU staining was performed using the Click-iT EdU Cell Proliferation Kit for Imaging (Thermo Fisher, MA) following the manufacturer's instructions. EdU (25 µg/fish) was IP injected 24 hr before extracting the heart. TUNEL assay was performed using In Situ Cell Death Detection Kit, TMR red (Sigma) following the manufacturer's instructions.

Imaging of whole-mount hearts and heart sections was performed using Nikon SMZ25 and Zeiss LSM 800, respectively. Proliferating CM and the density of CM nuclei in the 200 µm border zone directly adjacent to the injured area were quantified as previously described (*Marín-Juez et al., 2016*). Revascularization was examined by live imaging of the endogenous fluorescence from vessel reporter fish. Revascularized vessel density in the whole injured area was measured and quantified using ImageJ. Student's *t*-test was applied to assess all comparisons by Prism 9.

## In situ hybridization

In situ hybridization was performed on cryosections according to standard procedures. Briefly, the templates of antisense digoxigenin (DIG)-labeled riboprobes for *hbaa1*, *timp4.3*, and *hmox1a*

were generated by PCR with the additional T7 promoter sequence on the primers. Oligonucleotide sequences for probe synthesis are listed in the Key Resources Table. The PCR product was transcribed using DIG RNA labeling mix (Roche, Basel, Switzerland) and T7 RNA polymerase (Promega, WI). To prepare the cryosections, hearts were fixed in 4% PFA in diethyl pyrocarbonate (DEPC)-treated PBS at 4°C overnight and immersed in OCT as described in the section 'Cryosections and histological analyses.' Briefly, 18 um cryosections were used and dried at 37°C for 30 min, treated with 10 μg/ml proteinase K (Sigma) and post-fixed with 4% PFA for 15 min. Cryosections were prehybridized for at least 2 hr without probe and then hybridized with probes (2 ng/μl for *hbaa1* and 5 ng/μl for both *timp4.3* and *hmox1a*) at 65°C overnight. After serial washing with 1×SSC and 1×MABT, the cryosections were blocked for 1 hr at room temperature and then incubated with anti-Digoxigenin Fab fragments Antibody, AP Conjugated (1:1000; Roche, Basel, Switzerland) at 4°C overnight. The next day, the probes were detected by chromogenic reaction with NBT/BCIP stock solution (1:50; Roche) and observed by a slide scanner (Pannoramic 250 FALSH II).

## O-Dianisidine staining

O-dianisidine staining was performed to detect heme in sections of the injured hearts according to standard procedures. Briefly, cryosections were wash with PBST and incubated with the freshly prepared o-dianisidine solution (10 mM sodium acetate pH 4.5, 40% ethanol, 50 μM o-dianisidine, and 0.65% $H_2O_2$) for 10 min. Imaging was performed using a Nikon SMZ25 microscope. Quantification of the red color developed in the injured area was performed using ImageJ, and Student's *t*-test was applied for the statistics.

## Next-generation RNA sequencing analysis

For each time point and condition, zebrafish ventricles from three fish hearts were used as biological duplicates for the RNA-seq experiments. RNA extraction was performed as previously described with minor modifications (*Lai et al., 2017*). Briefly, RNA isolation was done using the miRNeasy micro Kit (QIAGEN). RNA quality analysis was done using Qubit and Bioanalyzer at the NGS High Throughput genomics core, Academia Sinica. Sequencing was performed on the HiSeq Rapid (Illumina) setup, resulting in a yield of 15–20 M reads per library on a 2 × 150 bp paired-end setup at the NGS core facility. Raw reads were assessed after quality control (QC), and output to filtered reads that were mapped to zebrafish Ensembl genome assembly using HISAT2 (*Kim et al., 2019*). The number of reads was aligned to Ensembl annotation using StringTie (*Pertea et al., 2015*) for calculating gene expression as raw read counts. The raw counts of the mapped annotated genes were joined to a combined matrix and normalized using upper-quantile normalization to generate normalized FPKM. Further analysis was done based on the normalized read values and respective log2 fold change to generate DEGs using NOIseq (*Tarazona et al., 2015*). GO enrichment analysis was performed in WebGestalt (*Zhang et al., 2005*) for the dataset considering the upregulated and downregulated genes against control versus treatment conditions and respective uninjured samples. Over-representation analysis was done with id of mapped genes for multiple testing corrections using Benjamini and Hochberg FDR correction and conducted with a significance level of 0.05. PCA was performed on normalized FPKM values of all the datasets at respective time points to analyze the sample level progression through time under control versus treatment conditions. Pathway analysis was performed using IPA software (QIAGEN; *Krämer et al., 2014*) following the manufacturer's protocols. Log2 ratio was input from the normalized read counts in zebrafish and defined as DEGs at log2FC above ±1.

## scRNAseq and bioinformatic analysis

Heart dissociation followed the same procedures in qPCR. For each scRNA-seq sample, 35–45 cryoinjured ventricles were collected from each experiment at respective timepoints and conditions, pooled together, and digested using LiberaseDH (Sigma) at 28°C in a 24-well plate. Cell suspension was filtered through 100 μm, 70 μm, and 40 μm cell strainers (SPL, Gyeonggi-do) and centrifuged at 200 × *g* for 5 min. Cell pellet was resuspended in 0.04% BSA/PBS, stained with DAPI and filtered through 35 μm Flowmi cell strainer (BD). Cells were sorted by FACS (BD Facs Aria) and subjected to scRNAseq following cell counting with countess II automated cell counter (Invitrogen).

scRNAseq library was generated using single cell 3' Reagent kits in chromium platform (10x Genomics). Cell ranger software suite (10x Genomics) was utilized for processing and de-multiplexing

raw sequencing data (*Zheng et al., 2017*). The raw base files were converted into the fastq format, and the subsequent sequences were mapped to zebrafish Ensembl genome assembly for processing. Downstream analysis of the gene count matrix generated by CellRanger (10x Genomics) was performed in R version of Seurat 3.1 (*Butler et al., 2018*; *Stuart et al., 2019*). The raw gene count matrix was loaded into Seurat, and a Seurat object was generated by filtering cells that expressed >400 nUMI counts and >200 genes. Additional filters for extra quality control were applied by filtering cells with log10 genes per UMI >0.8 and cells with mitochondrial gene content ratio lower than 0.23. This resulted in 9437 cells for the uninjured dataset, 16,657 cells for the PBS1d dataset, 9997 cells for PBS3d dataset, 11,950 cells for PBS7d dataset, 9912 cells for CL1d dataset, 12,621 cells for the CL3d dataset, and 11,236 cells for the CL7d dataset. Reads were normalized by the 'NormalizeData' function that normalizes gene expression levels for each cell by the total expression. The top 3000 highly variable genes were used as variable features for downstream analysis. Prior to dimensionality reduction, a linear transformation was performed on the normalized data. Unwanted cell–cell variation was eliminated by 'regress out' using mitochondrial gene expression during scaling. The datasets were integrated into a single Seurat object using the canonical correlation analysis (CCA) with an 'SCT' normalization method. Dimensionality reduction was performed on the integrated dataset using PCA. The top 40 principal components were identified based on PCElbowPlot. UMAP dimensional reduction by 'RunUMAP' function was used to visualize the cell clusters across conditions (*Becht et al., 2018*; *Figure 2—source data 1*). A total of 19 cell clusters were generated by 'FindClusters' function using 40 PCs and a resolution of 0.4. Following clustering, DEGs in each of the clusters were determined using 'FindMarkers' and 'FindAllMarkers' assessed by the statistical MAST framework (*Finak et al., 2015*). Top DEGs of each cluster were then filtered based on being detected in ≥25% of cells compared to other cells within the dataset, with a Bonferroni adjusted p-value<0.05. Clusters were assigned with specific cell identities based on the lists of DEGs and ordered by average log fold change and p-value. Visualization of specific gene expression patterns across groups on UMAP and heatmaps (top enriched genes) was performed using functions within the Seurat package. Gene enrichment analysis was determined using the above-described RNA-seq pipeline. In brief, the DEGs across cell types were used in WebGestalt to generate the cluster-specific biological processes of GO and KEGG pathways (*Raudvere et al., 2019*).

In addition to basic scRNA-seq analysis steps, cell-cycle scoring and regression were also performed. Zebrafish's cell-cycle-related orthologs to human were identified based on Ensembl annotation. All cells were assigned a cell-cycle score by 'CellCycleScoring' function. Effects of cell-cycle heterogeneity were regressed out by 'ScaleData' function with cell-cycle scores. Dimensionality reduction of PCA and UMAP, and clustering were performed with the same functions and parameters in the basic analysis steps (*Figure 2—source data 1*).

## Pseudobulk analysis

We generated reference matrixes of the unsupervised and un-normalized mean counts computed for each gene across all individual cells from each cell type using the Seurat and the SingleCellExperiment package (*Amezquita et al., 2020*). The raw matrices were split by cell type and the associated conditions of zebrafish (PBS-treated vs. CL-treated). Normalization was performed on the sparse gene matrix using total read counts and the total cell numbers for each cell type. The normalized count matrix was applied for differential expression analysis using NOISeq (*Tarazona et al., 2015*). Furthermore, average expression of genes across all the cell types detected by the scRNA-seq data from each time point was calculated using Seurat function 'AverageExpression,' which was used as a reference dataset for query. The hierarchical heatmap was generated by Morpheus (https://software.broadinstitute.org/morpheus).

## Ligand–receptor interaction network analysis

The zebrafish ligand–receptor pairs were derived from human ligand–receptor pair database as described previously (*Ramilowski et al., 2015*). We mapped the ligand–receptor pair orthologous to the human database and generated the fish database for use in interaction network. We kept the ligand–receptor pairs that showed expression in upper-quantile normalization as the cutoff to define the expressed data. Next, the macrophages and neutrophils that expressed the genes of ligands and receptors were determined based on the counts from pseudobulk analysis with an upper-quantile

expression at the given time point. The total number of the potential ligand–receptor interaction events between each cluster of macrophages and neutrophils was visualized using Cytoscape (*Shannon et al., 2003*).

## Pseudotemporal trajectory analysis

We adopted the macrophage and neutrophil populations from the datasets respectively after importing the Seurat object using the as.cell_data_set and chooseCells function in Monocle3 (*Cao et al., 2019*; *Trapnell et al., 2014*). UMAP dimensional reduction was applied to project the cells using the plotCells and clusterCells function. To learn the pseudotemporal trajectory, we then used the learnGraph and orderCells functions using modified parameters. We determined the DEGs over pseudotime using the topMarkers function. The top 3 DEGs were selected by greater than 0.1 fraction of cells expressing the given genes and plotted by plot_genes_by_group function. The following gene expression of the cell types was also plotted as function of pseudotime to generate cell trajectory analysis across pseudotime using the plot_genes_in_pseudotime function in Monocle3 (*Figure 6— source data 1*).

## Acknowledgements

We thank core facilities at the Institute of Biomedical Sciences, including Light Microscopy, Pathology, and DNA Sequencing Core; Dr. Meiyeh Jade Lu and the High Throughput Genomics Core at Biodiversity Research Center for NGS work; the Innovative Instrument Project (AS-CFII-111-212) for cell sorting service at Academia Sinica. We also thank all members of the Lai group for their valuable suggestions and Drs. Michele Marass and Arica Beisaw for the editorial consultancy. Research in Lai group has been funded by the Clinical Research Collaboration Grant from the Institute of Biomedical Sciences (IBMS-CRC108-P01), the Research Project Grant from the Ministry of Science and Technology (MOST 108-2320-B-001-032-MY2), and the Grand Challenge Project from the Academia Sinica (AS-GC-110-P7).

## Additional information

### Funding

| Funder | Grant reference number | Author |
|--------|------------------------|--------|
| Academia Sinica | IBMS-CRC108-P01 | Shih-Lei (Ben) Lai |
| Ministry of Science and Technology, Taiwan | MOST 108-2320-B-001-032-MY2 | Shih-Lei (Ben) Lai |
| Academia Sinica | AS-GC-110-P7 | Shih-Lei (Ben) Lai |

The funders had no role in study design, data collection and interpretation, or the decision to submit the work for publication.

### Author contributions

Ke-Hsuan Wei, Data curation, Formal analysis, Investigation, Visualization, Writing – original draft, Writing – review and editing; I-Ting Lin, Data curation, Formal analysis, Supervision, Investigation, Writing – original draft, Writing – review and editing; Kaushik Chowdhury, Tai-Ming Ko, Data curation, Formal analysis, Investigation, Visualization; Khai Lone Lim, Investigation, Visualization, Writing – review and editing; Kuan-Ting Liu, Data curation, Formal analysis; Yao-Ming Chang, Data curation, Software, Formal analysis, Visualization; Kai-Chien Yang, Conceptualization, Resources, Funding acquisition; Shih-Lei (Ben) Lai, Conceptualization, Resources, Data curation, Supervision, Funding acquisition, Project administration, Writing – review and editing

### Author ORCIDs

Ke-Hsuan Wei (ID) http://orcid.org/0009-0005-8935-8432
Shih-Lei (Ben) Lai (ID) https://orcid.org/0000-0002-1409-4701

### Ethics

This study was performed in strict accordance with the recommendations in the Guide for the Care and Use of Laboratory Animals of the Academia Sinica. All of the animals were handled according to approved institutional animal care and use committee (IACUC) protocols (Protocol ID: 18-12-1241) of Academia Sinica. The protocol was approved by the Committee on the Ethics of Animal Experiments of the Academia Sinica. All surgery was performed under tricaine anesthesia, and every effort was made to minimize suffering.

### Decision letter and Author response

Decision letter https://doi.org/10.7554/eLife.84679.sa1
Author response https://doi.org/10.7554/eLife.84679.sa2

## Additional files

### Supplementary files

• MDAR checklist

### Data availability

All data generated or analyzed during this study have been included in the manuscript and uploaded to a public entry for raw reads of both bulk and single-cell RNAseq on NCBI SRA database (accession no. PRJNA900299). Source data files of respective figures have been provided.

The following dataset was generated:

| Author(s) | Year | Dataset title | Dataset URL | Database and Identifier |
|---|---|---|---|---|
| Wei K, Chang Y, Lai S | 2022 | Comparative single-cell profiling reveals distinct cardiac resident macrophages essential for zebrafish heart regeneration | http://www.ncbi.nlm.nih.gov/bioproject/?term=PRJNA900299 | NCBI BioProject, PRJNA900299 |

The following previously published dataset was used:

| Author(s) | Year | Dataset title | Dataset URL | Database and Identifier |
|---|---|---|---|---|
| Lai S, Marín-Juez R, Moura P, Kuenne C, Lai KH, Tsedeke AT, Guenther S, Looso M, Stainier DY | 2017 | Comparative transcriptome profiling of zebrafish and medaka hearts following cardiac cryoinjury | https://zfin.org/ZDB-GENO-960809-7 | Zfin, ZDB-GENO-960809-7 |

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
