## [Editor Report]

The authors analyze changes in the gene expression of different immune cells during heart regeneration using single-cell RNA-sequencing and assess changes upon drug treatment that depletes macrophages. They find that drug treatment affects the gene expression profiles and abundance of different immune cells. The work provides a useful resource of gene expression data and a nice analysis supporting immune cell interactions during heart regeneration.

---

## [Decision Letter]

**Decision letter after peer review:**

Thank you for submitting your article "Comparative single-cell profiling reveals distinct cardiac resident macrophages essential for zebrafish heart regeneration" for consideration by *eLife*. Your article has been reviewed by 3 peer reviewers, and the evaluation has been overseen by a Reviewing Editor and Marianne Bronner as the Senior Editor. The reviewers have opted to remain anonymous.

Essential revisions:

1. For revision at the experimental level. the points outlined by the reviewers below would be a good addition

2. Some further bioinformatics analysis should be performed as suggested.

3. The manuscript requires rewriting/reformulating/putting into context.

*Reviewer #1 (Recommendations for the authors):*

The authors define macrophage-depleted zebrafish as "non-regenerating". I don't think is quite accurate. While it is true that these zebrafish do not regenerate as efficiently as those without prior clodronate depletion of macrophages, these zebrafish should more be more accurately as "macrophage depleted" rather than "non-regenerating". The regeneration defect is secondary to macrophage depletion and changes in gene expression do not necessarily reflect the regeneration defect as much as the loss of resident macrophages, which is true even after the number of circulation macrophages has recovered.

The discussion of all the gene expression data in the Results section would likely benefit from some streamlining and focus on more general principles. As it is currently written, it has a lot of detail, which isn't bad necessarily but it makes reading large sections of the Results section quite cumbersome.

*Reviewer #2 (Recommendations for the authors):*

Overall, we think that this paper provides useful information regarding the heterogeneity and transcriptomic behaviour of resident macrophages and neutrophils during regeneration. Regarding the second part of the paper, the pre-depletion of macrophages and assessment of regeneration, the work seems very derivative of the last author work. Although in the current paper, the authors use different timepoint to deplete the population and by that add the information that even very early depletion is sufficient to detrimentally affect regeneration, they do not expand on this point and instead analyse it in the same scope they did in the previous paper. The early pre-depletion result is very surprising as the expectation from resident macrophage (based on studies in mice) is to hold a self-maintaining ability which given sufficient time would partly replenish itself and could help regenerate the heart. In our view focusing more on what leads to a lack of regeneration in this model, would be of great interest.

One of the caveats of this work is that it draws biological conclusions solely based on transcriptomic data. Although this is common practice when working with scRNA seq datasets, a large part of the inferred conclusions could be tested by adding a dimension of "biological certainty" to otherwise suggestive work, however, we believe that this is outside of the scope of this current study.

As a general note, this paper is written in an understandable way, however, the level of details provided in the text for the majority of aspects such as the gene ontology enrichment (Figure 1d, line 261 "At 7 dpci, up-regulated DEGs in both PBS- and CL^-^treated hearts were associated to "immune/inflammatory response", "chemokine-mediated signaling", "proteolysis", "neutrophil chemotaxis", "phagocytosis", "cytokine-cytokine receptor interaction", and "NOD-like receptor signaling pathway" (7d-Common, Figure 1D and 1E). These processes correspond nicely with known repairing processes of immune cell infiltration and debris clearance in the first week of cardiac repair.") makes the reading of the paper challenging. We would suggest writing in a more concise way while attempting to keep the details (we acknowledge it is not an easy task).

*Reviewer #3 (Recommendations for the authors):*

1. I was missing some functional validations of some of the hub genes defining some of the relevant populations. One could though argue that this goes beyond the current article, which already counts with 8 Figures.

2. I was missing a description of Figure 8C (effect on cardiomyocyte proliferation) in the Results section. Also, a longer discussion of this in my point of view very unexpected results is missing. Would it mean that circulating macrophages are important for cardiomyocyte proliferation?

3. As mentioned in the introduction, previous reports describe subpopulations of macrophages involved in zebrafish heart regeneration (e.g. Ma et al. Bevan et al., Sanz-Morejon et al). The marker genes used in those studies are not put into context with the currently presented data in full. For example, which are the populations expressing osteopontin (ssp1) or wt1b?

4. IP injection of chlodronate leads to a reduction of macrophages in the heart. It might however also lead to macrophage ablation elsewhere in the body. Can the authors exclude that the reason why there is a lack of proper regeneration is that a chlodronate-sensitive macrophage population was ablated throughout the organisms, not only in the heart? If not, this possibility should be discussed.

5. It would be interesting to know if the phenotype is a delay in regeneration or a complete abrogation. Could the authors include long-term AFOG stainings, e.g. 100 dpci?

---

## [Author Response]

Essential revisions:Reviewer #1 (Recommendations for the authors):The authors define macrophage-depleted zebrafish as "non-regenerating". I don't think is quite accurate. While it is true that these zebrafish do not regenerate as efficiently as those without prior clodronate depletion of macrophages, these zebrafish should more be more accurately as "macrophage depleted" rather than "non-regenerating". The regeneration defect is secondary to macrophage depletion and changes in gene expression do not necessarily reflect the regeneration defect as much as the loss of resident macrophages, which is true even after the number of circulation macrophages has recovered.

We thank the reviewer for bringing up this issue and totally agree with this comment. We now define the models we used to compare and characterize macrophages more clearly as the macrophage-delayed model (-1d_CL treatment) for transcriptomic profiling and resident macrophage-deficient model for functional characterizations (-8d_CL treatment, without affecting monocyte-derived/circulating macrophages) throughout the manuscript.

The discussion of all the gene expression data in the Results section would likely benefit from some streamlining and focus on more general principles. As it is currently written, it has a lot of detail, which isn't bad necessarily but it makes reading large sections of the Results section quite cumbersome.

We thank the reviewer’s comment and have now streamlined the story and focus more on the general principles:

– Following our previous work as a research advancement, we compare the macrophage property and their interaction with neutrophils between -1d_CL^-^treated hearts and PBS-control hearts.

– Bulk-RNAseq revealed chronic inflammation and misregulated ROS homeostasis and metabolism upon -1d_CL treatment.

– Comparative scRNAseq profiling identified resident macrophage subsets are largely diminished in -1d_CL hearts while the recruited macrophages (mainly Mac 1) showed alternative polarization toward inflammatory (M1) function in -1d_CL hearts, instead of tissue-repairing functions in -1d_PBS control hearts.

– Cellular crosstalk analysis showed that the interactions between macrophage and neutrophils shifted from phagocytic neutrophil clearance and ECM remodeling under PBS condition to neutrophil retention and proinflammatory M1 signaling under -1d_CL condition. These results correspond nicely to the neutrophil retention and chronic inflammation phenotypes.

– To dissect the function of resident macrophage without affecting circulating monocyte/macrophage recruitment, we established and verified the resident macrophage-deficient model (-8d_CL). Functional analyses suggest resident macrophages are essential for heart regeneration, in terms of revascularization, CM survival, and scar resolution.

– Mechanistic study (during the revision) identified *homx1a* enriched in Mac 2 and functions in heme metabolism, potentially reducing the secondary oxidative damages of CMs and nicely corresponding to the CM survival defect.

– Collaborating with Dr. Arica Beisaw at the Institute of Experimental Cardiology, University of Heidelberg, we also found that CM protrusion and collagen denaturing are essential processes for the regenerated CMs to migrate and replace fibrotic scar. This process was abolished in the -8d_CL heart, suggesting that resident macrophages function in regulating collagen deposition and denaturing, in turn allowing CMs to protrude and replenish the infarct. These data will be incorporated into Dr. Beisaw’s manuscript in preparation.

– Taken together, we identified unique resident macrophage subsets prerequisite and non-recoverable for heart regeneration, potentially via modulating the infarct microenvironment.

Reviewer #2 (Recommendations for the authors):Overall, we think that this paper provides useful information regarding the heterogeneity and transcriptomic behaviour of resident macrophages and neutrophils during regeneration. Regarding the second part of the paper, the pre-depletion of macrophages and assessment of regeneration, the work seems very derivative of the last author work. Although in the current paper, the authors use different timepoint to deplete the population and by that add the information that even very early depletion is sufficient to detrimentally affect regeneration, they do not expand on this point and instead analyse it in the same scope they did in the previous paper. The early pre-depletion result is very surprising as the expectation from resident macrophage (based on studies in mice) is to hold a self-maintaining ability which given sufficient time would partly replenish itself and could help regenerate the heart. In our view focusing more on what leads to a lack of regeneration in this model, would be of great interest.One of the caveats of this work is that it draws biological conclusions solely based on transcriptomic data. Although this is common practice when working with scRNA seq datasets, a large part of the inferred conclusions could be tested by adding a dimension of "biological certainty" to otherwise suggestive work, however, we believe that this is outside of the scope of this current study.As a general note, this paper is written in an understandable way, however, the level of details provided in the text for the majority of aspects such as the gene ontology enrichment (Figure 1d, line 261 "At 7 dpci, up-regulated DEGs in both PBS- and CL^-^treated hearts were associated to "immune/inflammatory response", "chemokine-mediated signaling", "proteolysis", "neutrophil chemotaxis", "phagocytosis", "cytokine-cytokine receptor interaction", and "NOD-like receptor signaling pathway" (7d-Common, Figure 1D and 1E). These processes correspond nicely with known repairing processes of immune cell infiltration and debris clearance in the first week of cardiac repair.") makes the reading of the paper challenging. We would suggest writing in a more concise way while attempting to keep the details (we acknowledge it is not an easy task).

We really appreciate all the thoughtful comments and constructive suggestions! Indeed, the results from the early depletion model were very surprising to us, and we are indeed actively generating related tools to further investigate these residential populations as previously described.

Following the reviewer’s suggestion, we further investigate how resident macrophages may function in cardiac repair according to the hints from our profiling results. We first identified that *heme oxygenase 1a* (*homx1a*) highly enriched in Mac 2 and functions in heme metabolism. From the literature, heme released from damaged erythrocytes (hemoglobin), and in our context, CMs (myoglobin), are the major source of oxidative stress, which may cause extended cell damage (Reeder et al., Curr Med Chem. 2005). We further demonstrated that *hmox1a* is expressed in macrophages/*mpeg1*:mCherry+ cells in the PBS-control hearts by ISH and IHC, and were much reduced in -8d_CL hearts. In addition, we labeled heme by o-dianisidine and indeed found extensive accumulation in the -8d_CL hearts as an underlying mechanism of decreased CM survival (Figure 8).

In addition, we searched the literature for examples of failed regeneration with unaffected CM proliferation and found that CM protrusion and invasion may be a factor. Inspired by the unpublished work presented by my former colleague Dr. Arica Beisaw at the CPI meeting last year, we examined the collagen degradation/denaturing by collagen hybridizing peptide (CHP) staining and how this may be associated with CM protrusion in our resident macrophage-deficient model. As a result, we found stronger CHP signals around the protruding CMs with *mpeg1*:mCherry+ macrophages surrounding them in PBS-control hearts at 7 dpci, while both the protrusions and the CHP signals were significantly reduced in -8d_CL hearts, suggesting that resident macrophages function in ECM remodeling and CM protrusion. Together with Dr. Beisaw’s group, we are investigating the mechanism of how cardiac macrophages remodel ECM and interact with CMs during heart regeneration and preparing another manuscript on this topic.

**Author response image 1. sa2fig1:** Collagen degradation and CM protrusion were compromised in resident macrophage-deficient hearts. Denatured collagen and CM protrusions in the injury area were examined at 7 dpci under regenerative (-8d_PBS) or resident macrophage deficient condition (-8d_CL). The denatured collagen was labeled by collagen hybridizing peptide (CHP) staining. CMs were specified by *myl7*:DsRed2-NLS, while their cellular protrusions were labeled by phalloidin (arrows). White dotted lines delineated injury areas; scale bars, 100 μm (left panels) and 10 μm (right panels). CM, cardiomyocyte; CHP, Collagen Hybridizing Peptide. Quantification of CM protrusion (p=0.0128) and border zone CHP density (p=0.0411) was shown in the right panels. The heart samples under regenerative (-8d_PBS) or resident macrophage-deficient (-8d_CL) conditions are indicated by purple or green, respectively. The student’s t-test was used to assess all comparisons by Prism 9.

**Author response image 2. sa2fig2:** Resident macrophage-deficient hearts showed altered macrophage distribution and CM protrusion defects. Immunostaining of phalloidin, CHP and GFP for thick sections from 7-dpci hearts of *Tg(mpeg1:GFP)* under regenerative (-8d_PBS) or resident macrophage-deficient condition (-8d_CL). CM protrusion was depicted by phalloidin, colored in green, the degrading collagens were observed by CHP staining, colored in blue, and macrophages were colored in magenta; scale bars: 100 μm. The insets highlight the macrophage distribution with degraded collagen surrounding CM protrusion; scale bars: 10 μm. CM, cardiomyocyte; CHP, Collagen Hybridizing Peptide. The thickness of the sections: 20 μm.

**Author response image 3. sa2fig3:** Comparison of total collagen deposition between regenerative hearts and resident-macrophage deficient hearts. Quantification of total collagen of Masson Trichrome stain by ImageJ. Masson Trichrome was performed on the same slides of each heart after the CHP staining. The collagen stain was visualized by color deconvolution provided in ImageJ and was shown alongside the original image. Mask of collagen was obtained by setting a threshold in ImageJ and the collagen intensity in the injured area was calculated. The dotted lines delineated injury areas; scale bars: 100 μm. Quantification was shown in the right panel (p=0.0004). The heart samples under regenerative (-8d_PBS) or resident macrophage-deficient (-8d_CL) conditions are indicated by purple or green, respectively. The student’s t-test was used to assess the comparison by Prism 9. CHP, Collagen Hybridizing Peptide.

Last, we thank the reviewer for understanding the difficulty of describing the enormous data in a concise way while attempting to keep the details, as well as the scope of this current study. We have now rewritten the manuscript also according to reviewer #1’s suggestion and hope the reviewers find it much improved.

Reviewer #3 (Recommendations for the authors):1. I was missing some functional validations of some of the hub genes defining some of the relevant populations. One could though argue that this goes beyond the current article, which already counts with 8 Figures.

Following the reviewer’s suggestion, we have verified the cluster enriched genes *hbaa1* and *homx1a* in Mac2 and *timp4.3* in Mac 3, using multiple approaches including qPCR, ISH, and IHC (Figure 7—figure supplement 1C and D and Figure 8B and C). We further found heme accumulation in -8d_CL hearts as an outcome of reduced *hmox1a+* Mac2 directly associated with CM survival (Figure 8). We appreciate the reviewer’s understanding that some of the functional validations might go beyond the scope of the current manuscript.

2. I was missing a description of Figure 8C (effect on cardiomyocyte proliferation) in the Results section. Also, a longer discussion of this in my point of view very unexpected results is missing. Would it mean that circulating macrophages are important for cardiomyocyte proliferation?

Following the reviewer’s suggestion, we have described the missing part of the results in Figure 7—figure supplement 2C in the revised manuscript (p15, line 538) and further discussed this point in the Discussion section (p18, line 664).

3. As mentioned in the introduction, previous reports describe subpopulations of macrophages involved in zebrafish heart regeneration (e.g. Ma et al. Bevan et al., Sanz-Morejon et al). The marker genes used in those studies are not put into context with the currently presented data in full. For example, which are the populations expressing osteopontin (ssp1) or wt1b?

We were indeed familiar with those important studies and examined those genes in our datasets. However, we found no obvious correlation between those genes expression and our clustering, nor strong association with the regenerative condition in our models. Based on our data, macrophages express those genes sporadically in multiple clusters at moderate to low levels and in low cell numbers (see Author response image 4 for the data). Thus, we were cautious about emphasizing their potential significance in our manuscript.

**Author response image 4. sa2fig4:** Expression of *spp1* and *wt1b* among inflammatory cells in zebrafish hearts. (A) UMAPs and (B) violin plots of *spp1* and *wt1b* expression among inflammatory cell clusters.

4. IP injection of chlodronate leads to a reduction of macrophages in the heart. It might however also lead to macrophage ablation elsewhere in the body. Can the authors exclude that the reason why there is a lack of proper regeneration is that a chlodronate-sensitive macrophage population was ablated throughout the organisms, not only in the heart? If not, this possibility should be discussed.

We examined and found that the circulating macrophage recruitment and infiltration were not affected in -8d_CL hearts (Figure 7—figure supplement 1A). Still, we cannot exclude the possibility of resident macrophage deficiency in other organs (nor whether those will recover) or if depleting potential CL^-^sensitive hematopoietic niches might affect heart regeneration, as the reviewer pointed out. Thus, we have further discussed this possibility in the Discussion section following the reviewer’s suggestion (p18, line 677).

5. It would be interesting to know if the phenotype is a delay in regeneration or a complete abrogation. Could the authors include long-term AFOG stainings, e.g. 100 dpci?

We previously found that the scar tissue in those -1d_CL treated hearts did not recover at least 2 months post injury and exhibited very high mortality as time went by (Lai et al., 2017). In light of the mortality and the time frame of revision, it is difficult to examine whether the heart can regenerate after long-term recovery (e.g., 100 dpci). The debris clearance and the ECM remodeling/CM protrusion defects also suggest that the fundamental changes in the infarct microenvironment, which are unlikely to be compensated by the residual proliferating CMs even for a longer time.